# A resilient battery electric bus transit system configuration

Ahmed Foda [1] ✉, Moataz Mohamed [1] ✉, Hany Farag [2] & Ehab El-Saadany [3]

Electric mobility is fundamental to combat climate change and attaining the United Nations Sustainable Development Goals (SDG-11). However, electric mobility necessitates a seamless integration between power and transportation systems, as the resiliency of both systems is becoming far more interdependent. Here, we focus on disruption to Battery Electric Bus (BEB) transit system charging infrastructure and offer a resilient BEB transit system planning model. The proposed model optimizes the BEB system costs while ensuring the system's robustness against simultaneous charging station failures. In our case study, a single charging station failure would lead to up to 34.03% service reduction, and two simultaneous failures would reduce the service by up to 58.18%. Our proposed two-stage robust model addresses this issue with a relatively small added cost (3.26% and 8.12% higher than the base model). This cost enables uninterrupted BEB system operation during disruption, ensuring personal mobility, social interaction, and economic productivity.

The electrification of public transit is a promising solution to combat climate change[1]. Electric transit (e-Transit) renders substantial per-passenger greenhouse gas (GHG) emissions[2], aiding significantly in reducing the transportation sector's high GHG emissions share (20% of global $CO_2$ emissions)[3,4]. During the past decades, battery electric buses (BEBs) have been identified as a feasible alternative to fossil-fueled buses[5,6]. Moreover, BEBs' market share is growing rapidly (91.4% of the electric bus market in 2020) owing to their energy efficiency, quiet operation, low maintenance cost, and zero tailpipe emissions[7]. However, BEB transit system configuration requires advanced optimization models for charging infrastructure allocation, fleet configuration, energy management, and charging schedule[8–10]. Overall, the complexity of designing a BEB transit system stems from the need to balance contradictory objects/decisions, such as mitigating the upfront investment costs (e.g., charging stations and fleet), annual operational costs, GHG emissions, and utility grid impact. At the same time, the system should satisfy the operational timetable and level of service, energy supply considerations, and resiliency against uncertainties.

Several studies have considered en-route charging for BEBs, which exploits high-power fast chargers and utilizes layover times for charging, leading to uninterrupted operation[11,12]. En-route charging diminishes the need for a sizable battery size (overnight charging) or high labor cost (battery swapping)[13]. However, it relies on a seamless energy supply at the charging locations[14].

However, BEB transit system operation is susceptible to several disruptions, such as power outages and equipment malfunction (real-world examples are detailed in Supplementary Discussion 1). Recent studies show that current BEB optimization models are robust against disruptions if solved promptly (within 1 h)[15]. However, daily disruption of one charging station could reduce the frequency of the service by 57%. Therefore, unexpected disruptions can prevent BEBs from being effectively and timely charged, halting their operation.

Concerns about the effects of charging disruptions on the BEB system have been pointed out as a critical issue, and calls for plans, best practices, and models for a resilient BEB system are in place[16,17]. Charging station failure is a binary random variable, and it should be

[1]Department of Civil Engineering, McMaster University, L8S 8L4 Hamilton, ON, Canada. [2]Electrical Engineering and Computer Science Department, York University, M3J 1P3 Toronto, ON, Canada. [3]Electrical Engineering and Computer Science Department, Khalifa University, Abu Dhabi, United Arab Emirates. ✉e-mail: fodaa@mcmaster.ca; mmohame@mcmaster.ca

included in BEB system configuration planning models to achieve a resilient system design.

BEB transit systems are vulnerable to several external energy management disruptions, such as energy consumption rates, disruptions to charging components, and electricity outages. Recent studies focused on BEB system planning under uncertainty, as summarized in Supplementary Table 1. These studies developed BEB models incorporating uncertainty parameters related to energy consumption, travel time, charging time, passengers' boarding and alighting time, battery degradation, and power supply fluctuation[6,18,19].

Energy consumption is observed as the most investigated uncertain BEB operation parameter, as it is affected by numerous uncertain factors (e.g., the number of passengers, traffic, weather, etc.). Additionally, accurate energy consumption estimation is crucial to properly configure and operate the BEB system[20]. Some studies also investigated the uncertainty associated with the operation time[21].

Most relevant to the present study are the recent works of refs. 22, 23. Their work designed the BEB system considering power supply fluctuation uncertainty in single-stage robust optimization models using the budgeted uncertainty set. Their results, solidifying our argument, show that considering power supply fluctuation uncertainty impacts the resulting BEB system configuration depending on the priorities of decision-makers (changing the conservative level). Their charging power variability formulation differs from the charging station disruption uncertainty proposed herein.

First, the power supplied to a bus in a charging station was assumed to be a random variable that varies within a continuous uncertainty interval set. In comparison, the charging station failure should be formulated as discrete (whether the charging station is working or not). Second, the power supply fluctuation budgeted uncertainty set in their work is related to the bus lines/trips, not the charging stations[22,23]. In other words, the uncertainty set was formulated for each bus line in each trip and budgeted the summation of the supplied power deviation from the nominal value in all the charging stations in this bus line/trip.

That said, the scope of previous studies did not consider the optimal design of a resilient BEB system configuration under charging station disruption uncertainty, where the uncertainty set is formulated for the failure of the charging stations at the network level.

The definition of resiliency depends on application[24]. However, there is a consensus that systemic resiliency is the system's capacity to adjust or react to disruptive events[25]. Four different concepts of resiliency are coined[26]: (1) robustness: effectively managing disruptive events so that they have minimal or no impact on the functionality of the network. (2) Rebound: effectively resuming normal network operations after disruptions. (3) Extensibility: effectively addressing unexpected events that may disrupt current activities by extending system performance or capabilities. (4) Adaptability: effectively balancing competing priorities to develop the adaptive capacity that can respond to constantly changing contexts[26]. This study focuses on the robustness of BEB systems as the goal is to design a resilient BEB transit network against charging station disruption uncertainty.

Optimization models for network resiliency are introduced to assess how the network performance changes due to disruption and/or identify operation recommendations to overcome network fragility and vulnerability. The output decisions could be taken before (pre-perturbation), during, and/or after (post-perturbation) disruption.

Generally, models to improve network robustness include survivable network design, fault tolerance problems, two-stage stochastic network optimization, interdiction, $N - K$, attacker-defender, defender-attacker-defender, and robust optimization (RO), as detailed in ref. 25.

For BEB networks, charging station disruption conforms to removing network components (nodes/edges). The design of resilient networks that are robust against component failure uncertainty is often referred to as survivable network design models (pre-perturbation)[27]. These models ensure the resultant network is robust against up to $k$ network components removal while maintaining the network operation[25]. The survivable network design could be addressed in various ways, such as through probability-based models[28], RO models[29], and two-stage RO models[30]. The selection of system parameters before component failures naturally leads to min-max-min formulations, which nicely correspond to two-stage robust optimization problems. Moreover, the two-stage RO formulation avoids the necessity of the component failure distribution (probability-based models) and provides a less conservative design (single-stage RO models)[31]. As such, two-stage RO models have been widely used in many applications to design a survivable network under component failure uncertainty[32].

Overall, the two-stage robust optimization method is a practical approach to address the uncertainty in situations like charging station failures in the BEB system configuration planning. The focus is not on the probability of charging station failures but on ensuring that the BEB system works even in the worst-case scenario. Herein, system disruption is defined as a charging station failure that impacts the entire operation day.

This work develops a resilient BEB system configuration model that optimizes the total costs (capital and operational) while ensuring the model's robustness against $k$ simultaneous failures of charging stations, where $k$ is a conservative risk level supplied by the decision-maker. The proposed model provides the optimal BEB system infrastructure, including locations of the charging stations, stations' configurations (power of charger and the number of poles), and BEB's fleet battery sizes. Furthermore, it provides a resilient charging/operation schedule for the transit fleet under any $k$ charging station failures. The model is applied to a real-world, large-scale bus transit network and demonstrates its effectiveness. Undeniably, a resilient BEB system design comes with additional costs. The Price of Robustness of the Robust Models with $k = 1$ and $k = 2$ are marginal (3.26% and 8.12% higher than the Base Model, respectively). This cost prevents a significant service reduction if one (34%) or two (58%) charging stations fail.

## Results

### BEB system configuration: Base Model

The proposed resilient BEB system configuration model is formulated as a two-stage RO and applied to the transit network in Oakville City (see Supplementary Discussion 2). The solution framework is detailed in "Methods: Solution algorithm". In addition, the convergence of the solution framework is shown in Supplementary Discussion 3.

The BEB system configuration of the Base Model (no station failure) is presented in Table 1. The optimal BEB system under the nominal operation required five heterogeneous charging stations equipped with eight poles. A fleet of 91 BEBs with heterogeneous battery capacities is required to satisfy operation. Most buses (59.34%) are equipped with 100 kWh battery capacity.

**Table 1 | Results of BEB system configuration (Base Model)**

| | Total system cost ($/year) | BEB fleet configuration (# × kWh) | Number of charging stations | Power of charging units (# × kW) | Number of poles (#) |
|---|---|---|---|---|---|
| Base Model | $ 6,959,381.19 | 54 × 100<br>34 × 200<br>3 × 300 | 5 | 1 × 250<br>4 × 500 | 8 |

**Table 2 | System annual costs (Base Model)**

| Parameter | Amount ($) | Percentage (%) |
|---|---|---|
| Construction cost | $ 39,957.07 | 0.574% |
| Chargers cost | $ 83,004.14 | 1.193% |
| Battery cost | $ 697,916.73 | 10.028% |
| Fleet cost | $ 5,332,936.24 | 76.629% |
| Operational cost | $ 805,567.01 | 11.575% |
| Total annual cost | $ 6,959,381.19 | 100.00% |

**Table 3 | Failed buses and service reduction at r = 1 (Base Model)**

| Disrupted Station ID | 1 | 3 | 4 | 11 | 14 |
|---|---|---|---|---|---|
| Number of failed buses | 30 | 8 | 21 | 12 | 6 |
| Service reduction | 34.03% | 10.28% | 26.39% | 14.55% | 2.85% |

**Table 4 | Service reduction at r = 2 (Base Model)**

| Disrupted Station ID | 3 | 4 | 11 | 14 |
|---|---|---|---|---|
| 1 | 44.11% | 58.18% | 46.43% | 38.08% |
| 3 | | 38.72% | 22.93% | 14.59% |
| 4 | | | 44.54% | 30.12% |
| 11 | | | | 19.44% |

**Table 5 | Results of BEB system configuration (Robust Models)**

| Models | Total system cost ($/year) | BEB fleet configuration (# × kWh) | Number of charging stations | Power of charging units (# × kW) | Number of poles (#) |
|---|---|---|---|---|---|
| Base Model | $ 6,959,381.19 | 54 × 100<br>34 × 200<br>3 × 300 | 5 | 1 × 250<br>4 × 500 | 8 |
| Robust Model k = 1 | $ 7,186,845.54 | 54 × 100<br>33 × 200<br>2 × 300<br>1 × 500<br>1 × 600 | 18 | 9 × 250<br>9 × 500 | 19 |
| Robust Model k = 2 | $ 7,524,798.35 | 47 × 100<br>29 × 200<br>9 × 300<br>5 × 400<br>1 × 500 | 33 | 21 × 250<br>12 × 500 | 33 |

The distribution of the BEB system costs is presented in Table 2, with a total annual cost of $6,959,381.19. It should be highlighted that the capital expenses are amortized over the 12-year lifespan. The primary contributor is the fleet cost without batteries (76.63%), while operational costs (electricity ToU, demand charge, and GHG emissions) account for 11.58%. In comparison, BEBs' batteries contribute 10.03%, and the charging system (charging stations infrastructure and chargers) accounts for 1.77%.

The optimal five locations for the en-route charging stations are distributed across the network (see Supplementary Fig. 2). The number of BEBs charging at each station varies between 8 and 50 buses. In turn, the charging instances, overall duration of charging, and energy demand differ widely from one charging station to another. Each charging station failure scenario affects the BEB system operation differently. For example, Oakville GO Station (ID 1) is a critical charging point. It charges 50 BEBs out of 91, serving 15 routes, using 318 charging events with a total of 4837.02 kWh daily energy demand (27.45% of the fleet energy demand). Therefore, any disruption to this station is expected to significantly impact the system's operation.

The robustness of the Base Model is assessed by estimating the number of daily failed buses and service reduction in all the $r$ simultaneous charging station disruption scenarios ($r$ interdiction method). This robustness assessment is calculated by solving the inner subproblem (ISP) (Eqs. 75–77 in "Methods: Solution algorithm") for each failure scenario. It is worth noting that ISP is minimizing the impact of the charging station failure scenario on the BEB system operation by obtaining the optimal charging scheduling that reduces the number of failed buses. The results of $r = 1$ and $r = 2$ are presented in Tables 3 and 4, respectively.

In the event of a single charging station failure (designated as $r = 1$), the daily operation is reduced by up to 34.03%. In the case of two simultaneous charging station failures ($r = 2$), the failure of Stations ID 1 and 4 reduces the service by 58.18% due to a failure of 48 BEBs.

The robustness assessment of the Base Model emphasizes the sensitivity of the BEB system operation to the charging station failure. It demonstrates the importance of the developed resilient BEB system configuration model in preserving the full operation and level of service.

**A resilient BEB system configuration**
The BEB system configurations of the Robust Models ($k = 1$ & 2) are summarized in Table 5 (for more details, see Supplementary Tables 4–6). For the Robust Model ($k = 1$), the BEB charging system comprises 18 charging locations with a heterogeneous charger-rated power and 19 poles. Moreover, the fleet configuration differs from the Base Model, introducing BEBs with higher battery sizes (400 kWh, 500 kWh, and 600 kWh). This resulting BEB system configuration is robust against any single charging station failure. During failure, the model creates a new charging schedule utilizing the remaining charging stations and ensures that all BEBs complete their scheduled trips (an example is illustrated in Supplementary Tables 7 and 8). For instance, the energy demand of Bus ID 3 is 412.8 kWh drawn from four en-route charging stations ID 4, 7, 19, and 24 in the nominal operation (163.79 kWh, 33.33 kWh, 118.52 kWh, and 97.18 kWh, respectively). In the case of the failure of any of these four charging stations, Bus ID 3 charges the same amount from the remaining three stations (for more details, see Supplementary Tables 9 and 10).

In the Robust Model ($k = 1$), the day-to-day operational costs vary from $2,021.47 (no disruption) to $2,069.85 (disruption of Station ID 4). Supplementary Fig. 3 and Supplementary Table 11 illustrate the daily operational costs of all the scenarios of one and two charging station disruptions in the Robust Models with $k = 1$ and $k = 2$, respectively.

For a higher level of conservative ($k = 2$), a resilient BEB system requires 33 charging stations with 33 poles. The remaining 31 stations will satisfy the fleet energy demand if any two charging stations are jointly disrupted.

Even though the robust BEB system configuration has more allocated charging stations (18 locations in $k = 1$ and 33 locations in $k = 2$) compared to the Base Model (5 locations), the number of charging stations with multiple poles in the Robust Models is lower (one station in Robust Model with $k = 1$ and none in the Robust Model with $k = 2$). In comparison, the Base Model has two stations with multiple poles.

The energy demand shows the same trend (Supplementary Tables 12–14). The Base Model's highest daily energy demand location is 5679.78 kWh. However, in the robust BEB system configuration with $k = 1$ and $k = 2$, the highest daily energy demands are 2807.45 kWh (≈49% of the Base Model) and 1805.25 kWh (≈32% of the Base Model), respectively.

**The Price of Robustness (PoR)**
The proposed two-stage RO Model provides a resilient BEB system configuration against $k$ simultaneous charging stations. However, this

comes at an additional cost, as presented in Table 6. This cost is called the Price of Robustness (PoR) (see Eq. 22 in "Methods: Two-stage robust model").

The PoR of the robust BEB system configurations with $k = 1$ and $k = 2$ are 3.26% and 8.12%, respectively. Therefore, designing a robust BEB system for Oakville's transit network that can function with one charging station failure will only cost 3.26% more than the Base Model. This will prevent up to a 34.03% reduction in operation. The dollar value of such service reduction is enormous. For $k = 2$, a robust BEB system costs 8.12% more than the Base Model. However, it prevents up to 58.18% of service reduction if two charging stations are disrupted. Eventually, system robustness enhancement increases the total expenditure.

For further clarification, Fig. 1 shows the percentage cost deviation of each component between the Base Model and the Robust Models with $k = 1$ & 2. All the capital costs are increased in the Robust Models except the fleet cost (without batteries). The charging system costs are also increased, including infrastructure costs (number of stations) and chargers costs. The battery costs also increased, but not as much as the charging system costs. On the other hand, the operational costs in the Robust Models are lower. Similarly, the on-peak

### Table 6 | System annual costs (base and robust models)

| Parameter | Base Model | Robust Model, $k = 1$ | Robust Model, $k = 2$ |
|---|---|---|---|
| Construction cost | $ 39,957.07 | $ 143,845.43 | $ 263,716.63 |
| Chargers cost | $ 83,004.14 | $ 242,086.54 | $ 405,324.47 |
| Battery cost | $ 697,916.73 | $ 729,882.38 | $ 836,434.56 |
| Fleet cost | $ 5,332,936.24 | $ 5,332,936.24 | $ 5,332,936.24 |
| Capital costs | $ 6,153,814.18 | $ 6,448,750.60 | $ 6,838,411.89 |
| Operational costs[a] | $ 805,567.01 | $ 737,837.80 | $ 686,386.46 |
| Total annual cost | $ 6,959,381.19 | $ 7,186,588.40 | $ 7,524,798.35 |

[a]Operational costs include the electricity ToU, demand charges, and emissions costs and are estimated in the Robust Models based on the scenario of no charging station failures (nominal operation). Please note that the battery capacity of each model is a decision variable, hence the variation of the battery cost across models.

demand of the Robust Model with $k = 2$ is 77% lower than the Base Model. Most importantly, the total system GHG emissions of the Robust Model with k = 2 is 8% lower than that of the Base Model.

## Discussion

Several certain (e.g., maintenance) and uncertain actions (e.g., electricity outage and component failure) face BEB transit systems, accentuating the importance of a resilient BEB system configuration implementation. However, the cost of a resilient BEB system is related to the conservative level ($k$). The BEB system configuration, total system costs, and surely the degree of the system robustness depend on the $k$ value. The appropriate $k$ value for any given transit network should consider the project budget, the risk facing the charging system, the price of robustness (PoR) for several $k$ values, and the robustness assessment.

Motivated by these issues, this work contributes to the development of a two-stage RO model for a resilient BEB system configuration, planning, and operation that considers charging station disruptions. In the first stage, the model decides the optimal locations and configurations of the charging stations and the BEB fleet configuration based on the worst-case realizations in the charging station failures uncertainty set. In the second stage, the recourse decisions of the charging schedule are obtained based on the first-stage BEB system infrastructure and the revealed uncertainty. In summary, the main contributions of this work

1. The model estimates the optimal locations of the charging stations, the number of poles and charger-rated power (heterogeneous), and the BEB fleet battery capacities (heterogeneous). The model minimizes the BEB system infrastructure and operational costs while accounting for the electricity Time-of-Use (ToU) tariff and demand charges, the time-based spread of the GHG emissions intensity, and limit constraints of the utility grid demand.
2. The model guarantees that the optimal BEB system satisfies full operation under any charging station failure scenario while optimizing the BEB system configuration for planning and operation.
3. The proposed resilient BEB system configuration model is applied to a real-world, large-scale multiple hubs transit network under

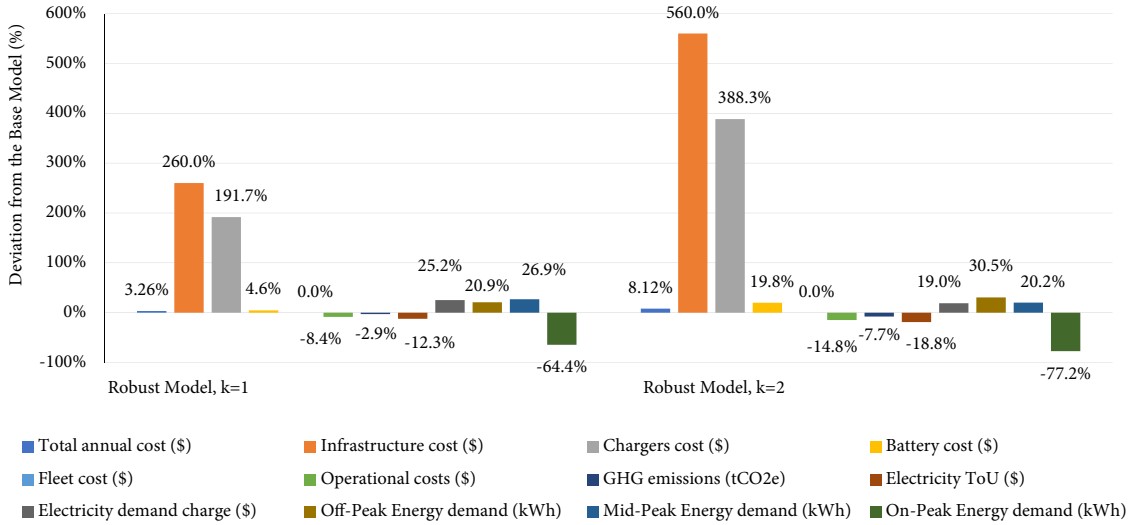

**Fig. 1 | Cost comparison between Base Model, Robust Model ($k = 1$), and Robust Model ($k = 2$).** Each parameter in the Robust Model with $k = 1$ (one charging station failure) and the Robust Model with $k = 2$ (two charging stations failure) is compared with the Base Model parameter (no charging station failure). These parameters include the system costs, GHG emissions, and energy demand in each period. Each bar represents the parameter's relative percentage change in Robust Models from the Base Model value. The relative percentage value is estimated using the following formula: (Robust Model parameter value − Base Model parameter value)/ (Base Model parameter value).

the nominal condition ($k = 0$) and different charging station failure budgets ($k = 1$ & $2$).
4. The model quantifies the price of robustness, enabling decision-makers to balance costs and risks.
5. The proposed resilient BEB system configuration model against charging station failure uncertainty could be readily implemented in practice.

Previous works in the literature, including our work in Foda[33], focused on developing static and generic BEB system configuration optimization models that are insensitive to the impact of charging station disruptions on the system configuration. Considering our work in Foda[33] and recent BEB literature, the BEB system design is not robust against charging station failure. Therefore, the service is susceptible to severe operation reduction during charging station failure.

Thus, in this study, the proposed model provides an optimal BEB system configuration design that satisfies full operation under any charging station failure scenario while optimizing the BEB system configuration for planning and operation. The obtained resilient BEB system design provides a feasible charging schedule under any $k$ simultaneous charging station failure scenario.

Specifically, this work is different from our previous work in Foda[33] across the following aspects: (1) the main objective here is to develop a resilient BEB system configuration that is robust against charging station disruptions. In comparison, the model in Foda[33] focused on providing a static and generic BEB system configuration for planning and charging schedules that did not consider the impact of the charging station disruption. (2) The optimization model in the present study is formulated as a two-stage robust optimization problem to address charging station failure uncertainty. The proposed model is solved using the nest column-and-constraint generation algorithm. In contrast, the model developed in Foda[33] is formulated as an integer linear programming problem and solved by an over-the-shelf solver. Therefore, the two models are different in the model structure.

Indeed, the BEB transit system is susceptible to unexpected disruptions that could prevent BEBs from charging. In turn, BEB system configuration predominantly relies on cost-minimization-based models, which are vulnerable to charging station disruptions[15]. Post-perturbation efforts (e.g., external generators or bus replacement) could either decrease the operation level of service (e.g., BEBs violate the timetable) or increase the costs, lowering the BEB system's economic and reliability competitiveness. These concerns emphasize the need for a resilient BEB system configuration model (pre-perturbation) that could handle the charging station failures uncertainty while minimizing the system costs and, most importantly, ensuring full-service provision.

We developed a two-stage Robust Optimization model with a budgeted uncertainty set that minimizes the total system annual costs, including capital and operational costs. This formulation produces fewer conservative designs compared to a single-stage model and avoids the necessity of failure probability distribution required in stochastic models.

The model is applied to a real-world, large-scale bus transit network and demonstrates its effectiveness. The model is solved under two different levels of disruption (conservative $k = 1$ & $2$), as summarized in Table 7. Undeniably, a resilient BEB system design comes with additional costs. However, these costs will be lower than any unoptimized post-perturbation activities to handle the charging station failures impact. The PoR of the Robust Models with $k = 1$ and $k = 2$ are 3.26% and 8.12%, respectively. This additional cost prevents a significant service reduction if one (34%) or two (58%) charging stations fail. Such service reduction could lead to enormous losses in economic productivity, personal mobility, and social interaction. Interested readers are encouraged to see Supplementary Discussion 4 for more details.

The proposed model is applied to another transit network with a different network structure, operational timetable, and number of buses (Guelph City, Ontario, Canada). The detailed results are presented in Supplementary Discussion 5.

Sensitivity analyses are conducted to understand the impact of parameter variations, including the charging system costs (infrastructure and chargers), battery costs, and operational costs (electricity ToU, demand charges, and GHG emissions). The results of the sensitivity analysis are detailed in Supplementary Discussion 6.

## Methods

A BEB system comprises three components: a transit network, a fleet of BEBs, and charging infrastructure (stations and poles). As such, it could be abstracted as a directed graph $G(N,E)$. A set of nodes $N$ representing potential charging locations, a set of edges $E$ representing routes served by BEBs.

This study develops a two-stage robust model for a generalized resilient BEB system configuration. The proposed model optimizes the total annual BEB system cost while handling up to $k$ charging stations failure ($k \geq 1$). The model estimates the optimal locations of the charging stations, system components (number and location of charger stations/poles, their rated power, and BEBs onboard battery size), and optimal charging schedule (which station, charging time, charging duration, and charged energy). Moreover, the model considers partial and continuous charging strategies, electricity ToU tariffs and demand charges, WTT GHG emissions temporal distribution, and the utility grid power constraints at the station and network levels. The proposed model follows six assumptions as follows:

- The operational timetable of the transit network is satisfied[34].
- All BEBs begin their operation day at full charge[35].
- The battery capacities of the BEB fleet are heterogeneous[36,37].
- The number of stations, poles, and charger-rated power are heterogeneous[38,39].
- System disruption is defined as a charging station failure that will impact the entire operation day[15].
- No backup buses are utilized to fulfill the trips of the failed buses due to the high added cost.

The rationale of the last assumption is (1) the high cost of the BEB relative to the cost of the charging station, (2) the high number of failed BEBs due to daily charging station disruption (more than the

## Table 7 | Cost and behavior of the proposed resilient BEB models

| Model | Base Model | Robust Model, $k = 1$ | Robust Model, $k = 2$ |
|---|---|---|---|
| Total annual cost (budget) | $ 6,959,381.19 | $ 7,186,588.40 | $ 7,524,798.35 |
| Price of Robustness (PoR) | NA | 3.26% | 8.12% |
| Additional cost from the previous model with $k - 1$ | NA | $ 227,207.21 | $ 338,209.95 |
| Maximum service reduction if one station fails | 34.03% | 0.0% | 0.0% |
| Maximum service reduction if two stations fail | 58.18% | 13.73% | 0.0% |
| Maximum service reduction if three stations fail | 59.67% | 22.38% | 8.92% |

## Table 8 | Abbreviations and notations

| Abbreviation | Description | Abbreviation | Description |
|---|---|---|---|
| BEB | Battery electric bus | SP | Sub-problem |
| WTT | Well-to-tank | LB | Lower bound |
| GHG | Greenhouse gas | UB | Upper bound |
| ToU | Time-of-Use | IMP | Inner master problem |
| SoC | State of charge | ISP | Inner sub-problem |
| MILP | Mixed integer linear programming | ILB | Inner lower bound |
| C&CG | Column and constrained generation | IUB | Inner upper bound |
| NC&CG | Nested column and constrained generation | GTFS | General Transit Feed Specification |
| MP | Master problem | VKT | Vehicle kilometers traveled |
| **Sets** | **Description** | **Index** | **Description** |
| $I$ | Set of potential charging station locations | $i, i'$ | Potential charging station location index |
| $B$ | Set of buses | $b$ | bus index |
| $J_b$ | Set of segments accomplished by bus $b$ | $j$ | Segment index |
| $A^{st}$ | Set of power levels of the station charger unit | $t$ | Timeslot index |
| $A^{batt}$ | Set of battery capacity levels | $q$ | Failure scenario index |
| $R_{b,j,i}$ | Set of timeslots of the recovering time of bus $b$ after accomplished segment $j$ at the location $i$ | $r$ | Inner-level C&CG scenario index |
| $\Xi$ | Set of the uncertainty of charging station failure | $\omega$ | Demand measurement interval index |
| $\Omega$ | Set of daily demand measurement intervals | | |
| $T$ | Set of the timeslots within a workday | | |
| $T_\omega$ | Set of timeslots in demand interval $\omega$ | | |
| **Parameters** | **Description** | **Parameters** | **Description** |
| $\rho^{st}$ | Construction cost of a charging station ($ per location) | $Q^{max}$ | Maximum battery size (kWh) |
| $\rho^{ch}$ | Charger cost ($ per kW) | $\lambda_1$ | Upper charging limit related to the battery capacity |
| $\rho^{po}$ | Charger pole cost ($ per unit) | $P_{i,t}^{max,st}$ | The station's upper limit of power in timeslot $t$ according to ToU (kW) |
| $\rho^{batt}$ | Battery cost ($ per kWh) | $\vartheta^{min}$ | Minimum SoC limit (%) |
| $\rho^{bus}$ | Bus cost without battery ($ per bus) | $\vartheta^{max}$ | Maximum SoC limit (%) |
| $\delta$ | Number of workdays (#) | $e^{base}$ | The constant energy consumption rate (kWh per km) |
| $T_s$ | Timeslot duration (hour) | $e^{batt}$ | The energy consumption rate factor relevant to the BEB battery size |
| $\rho_t^{elect}$ | Electricity rate according to ToU in timeslot $t$ ($ per kWh) | $P_{po}^{max}$ | The charger pole's maximum power (kW) |
| $\rho_t^{em}$ | WTT GHG emissions rate in the timeslot $t$ ($ per kWh) | $d_{b,j}$ | Segment $j \in J_b$ length of bus $b$ (km) |
| $\tau$ | Annualization factor (#) | $N_i^{max}$ | Maximum number of charger poles in charging station $i$ (#) |
| $\rho^{DC}$ | Electricity daily demand charges ($ per kW) | $k$ | Maximum number of failed stations (#) |
| $\eta^{ch}$ | Charger efficiency (%) | $\rho^{pen}$ | Penalty cost for the failed bus ($) |
| $P_t^{max,net}$ | The network upper limit of power in timeslot $t$ according to ToU (kW) | | |

spare ratio), (3) and the need to provide smooth operation to the passengers, drivers, and transit agency even during charging station disruption.

During operation, the recovering time between two consequence trips is exploited to charge BEBs. Specifically, during each recovering time, the decisions to charge and the charging duration are informed through the BEB system configuration optimization. Moreover, each charging event should satisfy two concepts: partial charging, which caps the charging duration by the available recovering time, and continuous charging, which restricts the charging to be in a continuous time interval. In addition, the charging power of each charging event is variable. It is related to several parameters, such as the charger pole-rated power limits, the available power depending on the station's charger unit-rated power, the number of BEBs charging simultaneously, and the BEB battery capacity.

For the reader's convenience, the abbreviations and notations are summarized in Table 8, and the decision variables are presented in Table 9.

### Base Model

The Base Model, without failure uncertainty of charging stations, is constructed as follows. Let $I$ denotes the set of candidates charging locations. The binary decision variable $x_i$ indicates whether location $i \in I$ is selected as a charging station. The selection of candidate charging station locations is based on three measures. First, the weighted degree centrality (the number of buses recovering) of each bus stop, terminal, or hub in the network is estimated, and stations with a higher weighted degree are prioritized. Second, for each route, the start, end, and any other en-route stop with a lengthy recovering time ($\geq 2\,T_s$) were considered. Third, the number of en-route candidate charging stations in the routes for each bus should be at least $k + 1$, where $k$ is the budget of the charging station failure. The rationale is to ensure that the resulting solution is flexible to address the failure uncertainty using all the available decisions (charging station location and configuration, onboard battery capacity, and charging schedule).

The charging station in location $i$ ($x_i = 1$) is equipped with a charger unit of rated power $P_i^{st}$ and a number of charger poles $N_i^{po}$. A penalty

**Table 9 | Decision variables**

| Decision variables | Description |
|---|---|
| $x_i$ | A binary decision variable denotes whether the location $i$ is chosen as a charging station or not |
| $N_i^{po}$ | A non-negative integer denotes the number of the charger poles in location $i \in I$ |
| $P_i^{st}$ | The charger unit-rated power at station $i \in I$ |
| $Q_b$ | The battery size for bus $b \in B$ |
| $z_{b,j}$ | A binary decision variable denotes whether the bus $b$ failed after serving segment $j$ or not |
| $y_{b,j,i,t}$ | A binary decision variable, $y_{b,j,i,t}$ = 1 if the bus $b$ charged in charging station $i$ after serving segment $j$ at timeslot $t$, $y_{b,j,i,t}$ = 0 otherwise |
| $\alpha_{b,j,i,t}$ | An auxiliary binary decision variable |
| $\gamma_{b,j,i,t}$ | An auxiliary binary decision variable |
| $P_{b,j,i,t}$ | A non-negative continuous variable indicates the charging-rated power for bus $b$ after accomplishing segment $j$ in location $i$ at timeslot $t$ |
| $P_i^d$ | Peak power demand of charging station $i$ |
| $P_{i,\omega}^{avg}$ | Average power demand in charging station $i$ during demand interval $\omega$ |
| $S_{b,j,i}^{dep}$ | A non-negative continuous variable presents the battery level of bus $b$ before departure to serve segment $j$ from the potential location $i$ (kWh) |
| $\xi_i$ | A binary decision variable, $\xi_i$ = 1 if the charging station in location $i$ is failed, $\xi_i$ = 0 otherwise |
| $\sigma$ | An auxiliary continuous variable |
| $\theta$ | An auxiliary continuous variable |
| $\pi$ | Dual variable for the sub-problem continuous variables |

cost of unsatisfied operation is added, which accounts for failed BEB trips during disruption. This is to meet the relative complete recourse property of the proposed two-stage robust model. For each bus $b \in B$, $z_{b,j}$ is a binary variable indicating if bus $b$ will fail to operate after serving segment $j$ due to insufficient battery state of charge (SoC) under the lower limit. A high penalty cost with a suitably large value is included ensuring that a full operation is satisfied. The trips of each bus $b$ are split into a set of segments $J_b$, where the index $j \in J_b$ indicates a sub-trip between two consecutive potential charging locations. The working day is discretized to uniform timeslots $T_s$ and is indexed by $t \in T$. The objective function of the Base Model in (1) aims to minimize the total annual BEB system cost, comprising six terms. The capital costs are annualized using a factor $\tau$ related to lifespan and the discount rate.

$$\min_{x,P^{st},N^{po},Q,z,y,\alpha,\gamma,P,P^d,P^{avg},S^{dep}} \tau \sum_{i \in I} \rho^{st} x_i + \tau \sum_{i \in I} \left( \rho^{ch} P_i^{st} + \rho^{po} N_i^{po} \right) + \tau \sum_{b \in B} \left( \rho^{batt} Q_b + \rho^{bus} \right)$$

$$+ \sum_{b \in B} \sum_{j \in J_b} \rho^{pen} z_{b,j} + \delta T_s$$

$$\left[ \sum_{t \in T} \sum_{i \in I} \sum_{b \in B} \sum_{j \in J_b} \left( \rho_t^{elect} + \rho_t^{em} \right) P_{b,j,i,t} \right] + \delta \sum_{i \in I} \rho^{DC} P_i^d$$

$$(1)$$

In Eq. (1), the construction cost of charging stations is ($\rho^{st}$), followed by the cost of charging components (charger units $\rho^{ch}$ and charger poles $\rho^{po}$). The fleet cost includes batteries $\rho^{batt}$ and buses $\rho^{bus}$. The penalty cost ($\rho^{pen}$) is allocated for unsatisfied trips due to failed buses if they exist. The operational costs include electricity ToU tariffs ($\rho_t^{elect}$), well-to-tank (WTT) GHG emissions ($\rho_t^{em}$), and demand charges ($\rho^{DC}$). The first two parts of the operational costs are related to the power to charge bus $b$ in station $i$ after serving segment $j$ at timeslot $t$ ($P_{b,j,i,t}$), and the latter is related to the daily peak power demand ($P_i^d$).

It is worth noting that including the temporal WTT GHG emissions resulting from electricity generation in the proposed model aims to decrease the overall GHG emissions of the entire system, enhances the environmental competitiveness of the BEB system, and aligns with carbon pollution pricing policies[40]. In Rupp[35], the charging schedule is optimized, incorporating the electricity ToU rates and the time-based values of CO$_2$e emissions per kWh. The research reveals that integrating WTT GHG emissions into the optimization process results in a

14.9% reduction in total system GHG emissions. Additionally, in Foda[33], when both the ToU of WTT GHG emissions and electricity tariffs are taken into account, the total annual WTT GHG emissions decrease by 13.34%.

The model in (1) is solved under a set of constraints (2–18). Constraints for the battery state of charge (SoC) are described in (2–5). The battery capacity of each bus $b$ during operation is limited to a predetermined range to satisfy the scheduled trips. In (2 and 3), the arrival battery capacity of bus $b$ to location $i$ after serving segment $j$ should be higher than a minimum value ($\vartheta^{min} Q_b$). Equation (2) restricts the arrival battery level for the first sub-trip as the buses are fully charged at the start of the operation $\vartheta^{max} Q_b$ (Assumption 2), while Eq. (3) constrains the remaining sub-trips. Where $S_{b,j,i}^{dep}$ is the battery level when bus $b$ departs from location $i$ to serve segment $j$ and $d_{b,j}(e^{base} + e^{batt} Q_b)$ is the consumed energy in this segment. The consumed energy during operation in segment $j$ is a function of the segment length $d_{b,j}$ and the energy consumption rate. In this work, the energy consumption rate is formulated as $(e^{base} + e^{batt} Q_b)$, where $e^{base}$ is the energy consumption rate for the base component without including the battery weight, $e^{batt} Q_b$ is the variable component of the energy consumption rate due to the weight of a battery pack with capacity $Q_b$, and $e^{batt}$ denotes the extra energy consumption rate resulting from an increase in the battery size by one unit[41,42]. It is worth noting that raising the bus battery capacity results in an increase in the weight of the battery pack, subsequently leading to a higher bus energy consumption rate[43]. While the departure battery capacity ($S_{b,j,i}^{dep}$) is restricted below a maximum percentage ($\vartheta^{max}$) of the battery capacity ($Q_b$) as mentioned in Eq. (4). Moreover, the relation between the departure battery capacity of bus $b$ from location $i$ to serve segment $j$ ($S_{b,j,i}^{dep}$) and the following segment departure battery capacity $S_{b,j+1,i'}^{dep}$ is presented in Eq. (5).

In the case of a bus $b$ failure after serving segment $j$ ($z_{b,j} = 1$), according to Eq. (5), all battery capacity constraints will be redundant, the bus will be out of the charging network, and a penalty cost will be added. While if no failure occurs, Eq. (5) emphasizes that the battery capacity when bus $b$ departure to segment $j + 1$ ($S_{b,j+1,i'}^{dep}$) equals the summation of the battery capacity when the bus departure to the previous segment $j$ ($S_{b,j,i}^{dep}$) and the charged energy during the recovering time after serving segment $j$ ($\sum_{t \in R_{b,j,i'}} \eta^{ch} T_s P_{b,j,i',t}$) minis the consumed

energy during segment $j$.

$$\vartheta^{\max} Q_b - d_{b,j}\left(e^{base} + e^{batt} Q_b\right) \geq \vartheta^{\min} Q_b \qquad \forall b \in B, j = 1 \qquad (2)$$

$$S_{b,j,i}^{dep} - d_{b,j}\left(e^{base} + e^{batt} Q_b\right) \geq \vartheta^{\min} Q_b \qquad \forall b \in B, \forall j \in J_b \backslash 1, i \in I \qquad (3)$$

$$S_{b,j,i}^{dep} \leq \vartheta^{\max} Q_b \qquad \forall b \in B, \forall j \in J_b \backslash 1, i \in I \qquad (4)$$

$$S_{b,j+1,i'}^{dep} \leq S_{b,j,i}^{dep} - d_{b,j}(e^{base} + e^{batt} Q_b) + \sum_{t \in R_{b,j,i'}} \eta^{ch} T_s P_{b,j,i',t}$$
$$+ Q^{\max} \sum_{j' \leq j} z_{b,j'} \qquad (5)$$
$$\forall b \in B, \forall j, j' \in J_b \backslash 1, i \& i' \in I$$

Through Eqs. (6 and 7), the charging power of bus $b$ after serving segment $j$ in location $j$ during timeslot $t$ ($P_{b,j,i,t}$) is bounded. Constraint (6) works in two ways. First, the charging power $P_{b,j,i,t}$ is zero if the binary decision variable of charging bus $b$ in timeslot $t$ ($y_{b,j,i,t}$) is zero. In addition, $P_{b,j,i,t}$ is lower than a maximum charger pole's power limit $P_{po}^{\max}$. In Eq. (7), the charging power is lower than a maximum factor ($\lambda_1$) multiplied by the battery capacity. This factor is related to the battery C-rate (the rate of charge or discharge relative to the battery capacity). Charging above the C-rate will lead to accelerated battery fading[44].

$$P_{b,j,i,t} \leq P_{po}^{\max} y_{b,j,i,t} \qquad \forall b \in B, \forall j \in J_b, i \in I, \forall t \in R_{b,j,i} \qquad (6)$$

$$P_{b,j,i,t} \leq \lambda_1 Q_b \qquad \forall b \in B, \forall j \in J_b, i \in I, \forall t \in R_{b,j,i} \qquad (7)$$

Equations (8–10) establish a continuous charging strategy using the auxiliary variables $\alpha_{b,j,i,t}$ and $\gamma_{b,j,i,t}$. From Eq. (8), $\alpha_{b,j,i,t}$ captures the status of unplugging the charger pole. Similarly, in Eq. (9), $\gamma_{b,j,i,t}$ equals one if the charger pole is plugged into the bus. Therefore, Eq. (10) enforces the charging continuity by setting the status change of plugging and unplugging to be less than or equal to one during the same charging process.

$$\alpha_{b,j,i,t} \geq y_{b,j,i,t} - y_{b,j,i,t+1} \qquad \forall b \in B, \forall j \in J_b, i \in I, \forall t \in R_{b,j,i} \qquad (8)$$

$$\gamma_{b,j,i,t} \geq y_{b,j,i,t} - y_{b,j,i,t-1} \qquad \forall b \in B, \forall j \in J_b, i \in I, \forall t \in R_{b,j,i} \qquad (9)$$

$$\sum_{t \in R_{b,j,i}} \alpha_{b,j,i,t} = \sum_{t \in R_{b,j,i}} \gamma_{b,j,i,t} \leq 1 \qquad \forall b \in B, \forall j \in J_b, i \in I, \forall t \in R_{b,j,i} \qquad (10)$$

The charging station constraints are presented in Eqs. (11–13). Equation (11) guarantees that if no charging station is built in location $i$, there is no charger unit with $N_i^{po}$ charger poles deployed at this location. Moreover, the number of charger poles utilized in any allocated station is bounded by an upper limit $N_i^{\max}$ which varies across locations. Equation (12) ensures that the number of buses charging in the same timeslot $t$ and location $i$ is less than or equal to the deployed charger poles. Similarly, in Eq. (13), the summation of the charging power at the same location and timeslot will not exceed the charger-rated power at this charging station ($P_i^{st}$). In Eqs. (14 and 15), the total charging power during each timeslot $t$ for each allocated station and network level is restricted to be less than the upper bounds that depend on the ToU.

$$N_i^{po} \leq N_i^{\max} x_i \qquad \forall i \in I \qquad (11)$$

$$\sum_{b \in B} y_{b,j,i,t} \leq N_i^{po} \qquad \forall i \in I, j, \in J_b, \forall t \in T \qquad (12)$$

$$\sum_{b \in B} P_{b,j,i,t} \leq P_i^{st} \qquad \forall i \in I, j \in J_b, \forall t \in T \qquad (13)$$

$$\sum_{b \in B} P_{b,j,i,t} \leq P_{i,t}^{\max,st} \qquad \forall i \in I, \forall t \in T \qquad (14)$$

$$\sum_{i \in I} \sum_{b \in B} P_{b,j,i,t} \leq P_t^{\max,net} \qquad \forall t \in T \qquad (15)$$

The electricity demand charges are determined by the peak power demand recorded during a billing period. It is measured as the maximum average electric power consumed within a specific time interval (e.g., 1 h). The average power of each demand measurement interval $\omega \in \Omega$ in station $i$ is calculated based on Eq. (16), where $\Omega$ is the set of daily demand charges measurement intervals and $T_\omega$ is the set of timeslots in demand interval $\omega$. In turn, in Eq. (17), the maximum daily power demand $P_i^d$ in charging station $i$ is estimated.

$$P_{i,\omega}^{avg} = \frac{\sum_{t \in T_\omega} \sum_{b \in B} P_{b,j,i,t}}{|T_\omega|} \qquad \forall i \in I, \forall \omega \in \Omega \qquad (16)$$

$$P_i^d \geq P_{i,\omega}^{avg} \qquad \forall i \in I, \forall \omega \in \Omega \qquad (17)$$

Equation (18) imposed the types of variables, such as $x_i, z_{b,j}, y_{b,j,i,t}, \alpha_{b,j,i,t}$ and $\gamma_{b,j,i,t}$ are binary, $N_i^{po}$ is a non-negative integer, and $P_{b,j,i,t}, P_i^d, P_{i,\omega}^{avg}$, and $S_{b,j,i}^{dep}$ are continuous. The variables $P_i^{st}$ and $Q_b$ are selected from predefined finite sets that represent a wide range of the available values/specifications in the market with the recent technology. The Base Model of the BEB system configuration is formulated as a mixed integer linear programming (MILP).

$$
\begin{aligned}
x_i &\in \{0,1\} & \forall i \in I \\
P_i^{st} &\in A^{st} & \forall i \in I \\
N_i^{po} &\in Z_{0+} & \forall i \in I \\
Q_b &\in A^{batt} & \forall b \in B \\
z_{b,j} &\in \{0,1\} & \forall b \in B, \forall j \in J_b \\
y_{b,j,i,t} &\in \{0,1\} & \forall b \in B, \forall j \in J_b, i \in I, \forall t \in R_{b,j,i} \\
\alpha_{b,j,i,t} &\in \{0,1\} & \forall b \in B, \forall j \in J_b, i \in I, \forall t \in R_{b,j,i} \\
\gamma_{b,j,i,t} &\in \{0,1\} & \forall b \in B, \forall j \in J_b, i \in I, \forall t \in R_{b,j,i} \\
P_{b,j,i,t} &\geq 0 & \forall b \in B, \forall j \in J_b, i \in I, \forall t \in R_{b,j,i} \\
P_i^d &\geq 0 & \forall i \in I \\
P_{i,\omega}^{avg} &\geq 0 & \forall i \in I, \forall \omega \in \Omega \\
S_{b,j,i}^{dep} &\geq 0 & \forall b \in B, \forall j \in J_b, i \in I
\end{aligned}
\qquad (18)
$$

## Two-stage robust model

The two-stage robust model satisfies all the charging station failure scenarios at the minimum total system costs. A budgeted uncertainty set presented in Eq. (19) is defined to allow up to $k$ simultaneous failures of charging stations from the beginning of the BEB system operation. Where $\xi_i$ is a random binary variable denoting whether the

charging station in location $i \in I$ fails.

$$\Xi = \left\{ \xi \in \{0,1\}^{|I|} : \sum_{i \in I} \xi_i \le k \right\} \tag{19}$$

Given the uncertainty set $\Xi$ defined in Eq. (19), the resilient BEB system configuration model is formulated as follows.

$$\min_{x, P^{st}, N^{po}, Q} \left[ \tau \sum_{i \in I} \rho^{st} x_i + \tau \sum_{i \in I} \left( \rho^{ch} P_i^{st} + \rho^{po} N_i^{po} \right) + \tau \sum_{b \in B} \left( \rho^{batt} Q_b + \rho^{bus} \right) \right.$$

$$+ \max_{\xi \in \Xi} \min_{z, y, \alpha, \gamma, P, P^d, P^{avg}, S^{dep}} \left[ \sum_{b \in B} \sum_{j \in J_b} \rho^{pen} z_{b,j} \right.$$

$$\left. \left. + \delta T_s \left( \sum_{t \in T} \sum_{i \in I} \sum_{b \in B} \sum_{j \in J_b} \left( \rho_t^{elect} + \rho_t^{em} \right) P_{b,j,i,t} \right) + \delta \sum_{i \in I} \rho^{DC} P_i^d \right] \right]$$

(20)

Subject to: (2–12), (14–19), and

$$\sum_{b \in B} P_{b,j,i,t} \le P_i^{st} \left( 1 - \xi_i \right) \qquad \forall i \in I, j \in J_b, \forall t \in T, \xi \in \Xi \tag{21}$$

In this two-stage RO model, the decisions of the charging stations allocation, charging unit-rated power, the number of charger poles in each station, and the battery capacity of each bus are taken in the first stage of minimization (here-and-now decisions). However, the actual operation decisions, such as the charging schedule (charging decision, charging power, and segments-specific departure battery SoC) and the bus failure decision, are taken in the second stage after the charging station perturbations have occurred via the maximization over the uncertainty set $\Xi$ (wait-and-see decisions).

Most notably, multiplying the daily operational costs in the objective function by the parameter $\delta$ (number of workdays) does not mean that the charging station failure $\xi \in \Xi$ will last for all the $\delta$ days. This approach is utilized as the operational costs are estimated daily (depending on the charging schedule), and the objective function is represented annually. Moreover, the two-stage optimization model relies on minimizing the impact of the worst-case scenario approach, which aligns with assuming the annual operational costs are caused by the worst-case charging station failure on all days. However, this is just the worst-case scenario. The actual annual operational costs of the obtained resilient model are estimated after the realization of each day charging station failure uncertainty.

The objective function of the two-stage RO model is solved under the same constraints of the Base Model after modifying Constraint (13) to Eq. (21) to include the uncertainty budget set constraint presented in Eq. (19) and have the failure decision and the associate uncertainty set in the model.

Equation (21) ensures that the charging power in the failed station is zero. Moreover, this emphasizes that the charging decision variable $y_{b,j,i,t}$ of bus $b \in B$ after serving segment $j$ during timeslot $t \in T$ in failed station $i$ will equal to zero as restricted in Eq. (6). It is worth noting that this modification does not restrict charging infrastructure allocation in the failed location. In contrast, and due to the maximization $\max_{\xi \in \Xi}$ to estimate the worst-case scenarios, the random disruption variable $\xi_i$ will equal to one only in locations $i$ chosen to build charging stations ($x_i = 1$). Toward this end, the proposed resilient model provides the BEB system infrastructure planning (number and configuration of charging stations and BEBs configuration) that is robust against the charging system infrastructure disruption.

The price of robustness is calculated using Eq. (22)[27], where ROSC(k) and DOSC are the optimal annual system cost of the Robust

Model with budget $k$ and the deterministic Basic Model. The PoR of the robust BEB system at each value $k$ is calculated by comparing the annual system cost with the Base Model ($k = 0$).

$$PoR(k) = \frac{ROSC(k) - DOSC}{DOSC} \tag{22}$$

## Solution algorithm

The proposed two-stage RO model is considered a tri-level optimization problem with mixed integer variables, which takes the form of $\min_{x, P^{st}, N^{po}, Q} - \max_{\xi \in \Xi} - \min_{Z, y, \alpha, \gamma, P, P^d, P^{avg}, S^{dep}}$. The solution algorithms are detailed in Supplementary Discussion 7. Moreover, computational performance enhancement approaches for the solution algorithm are described in Supplementary Discussion 8. Fig. 2 depicts a framework of the solution algorithm.

Nevertheless, the two-stage RO problems are generally NP-hard and computationally intractable for more realistic size models[45]. Moreover, the proposed model's second-stage problem is mixed-integer programming (i.e., $z, y, \alpha, \gamma \in \{0, 1\}$). Therefore, the strong duality cannot be directly applied to the second-stage problem $\max_{\xi \in \Xi} - \min_{Z, y, \alpha, \gamma, P, P^d, P^{avg}, S^{dep}}$. This condition renders the standard decomposition methods such as the L-shaped method[46], Benders decomposition[47], and classical column-and-constrained generation (C&CG) method[27,48] unsuitable to solve the proposed model. Therefore, we apply the nested column-and-constrained generation (NC&CG) framework developed by Zhao and Zeng[49] to solve the two-stage RO models with mixed-integer recourse problem. The NC&CG method is proven to be an exact algorithm for solving the two-stage RO problem in finite steps[49].

In the NC&CG algorithm, the two-stage RO model is decomposed into outer-level and inner-level problems. Each problem is solved using the standard C&CG method. Specifically, the outer-level problem calculates the BEB system infrastructure configuration by solving the first-stage problem under the worst-case scenarios estimated using the subproblem. While the inner-level problem iteratively solves the second-stage problem to identify the worst-case scenarios for the given BEB system infrastructure configuration (i.e., $x, P^{st}, N^{po}, Q$) resulted from the outer-level problem.

For any resulting first-stage variables ($x, P^{st}, N^{po}, Q$), the second-stage problem is feasible. As the recourse variables related to the charging schedule ($y, \alpha, \gamma, P, P^d, P^{avg}, S^{dep}$) and penalizing the unsatisfied operational trips are addressed in the objective function of the second stage ($z$). Therefore, the relative complete recourse condition is maintained in the proposed model under any scenario estimated according to the maximization function with respect to the random failure variable ($\xi$).

**Outer-level C&CG solution algorithm.** The outer level of the NC&CG algorithm is considered a standard C&CG framework[49]. Generally, the C&CG procedure comprises two problems, a Master Problem (MP) and a Sub-Problem (SP), solved in an iterative process[48]. This is based on the formulation of the two-stage RO problem that includes a master problem (first stage) in the form of min and a sub-problem (second stage) in the form of max − min.

The MP and the SP of the proposed two-stage RO BEB system configuration model are presented in Eqs. (23–40) and Eqs. (41–50), respectively.

[MP]:

$$\min_{x, P^{st}, N^{po}, Q, z^q, y^q, \alpha^q, \gamma^q, P^q, P^{d,q}, P^{avg,q}, S^{dep,q}} \tau \sum_{i \in I} \rho^{st} x_i + \tau \sum_{i \in I} (\rho^{ch} P_i^{st} + \rho^{po} N_i^{po}) + \tau \sum_{b \in B} (\rho^{batt} Q_b + \rho^{bus}) + \sigma$$

(23)

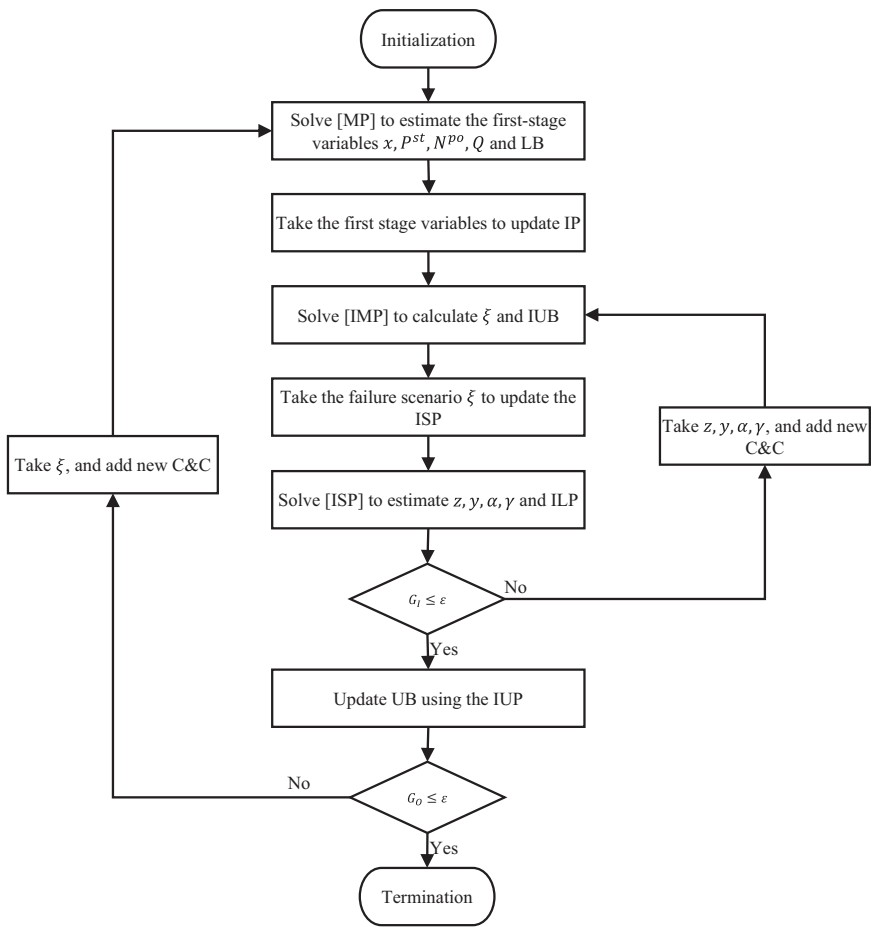

**Fig. 2 | Nested column-and-constraint generation solution algorithm.** The algorithm started with solving the master problem (MP) to estimate the first-stage variables (locations of the charging stations $x$, charger-rated power in each station $P^{st}$, number of poles $N^{po}$, and fleet battery capacities $Q$) and the iteration lower bound (LB). Then, the sub-problem (SP) is solved using the column-and-constraint (C&C) generation method. The SP includes solving the inner master problem (IMP) to obtain the failure scenario $\xi$ and the inner problem upper bound (IUB), along with solving the inner sub-problem (ISP) to get the value of the discrete variables of the ISP ($z,y,\alpha,\gamma$) and the inner problem lower bound (ILP). The values of the ISP

discrete variables $z,y,\alpha,\gamma$ are used iteratively to update the IMP until the convergence condition of the SP is reached (relative optimality gap of the inner problem $G_I$ is lower than a small number $\varepsilon$). The solution of the SP is taken as the iteration upper bound (UB). The obtained failure scenario from this iteration ($\xi$) is used to update the MP of the next iteration by adding the associated C&C. This procedure runs iteratively until the termination criteria of the outer problem are achieved (relative optimality gap of the outer problem $G_O$ is lower than a small number $\varepsilon$).

Subject to: (2), (11),

$$\sum_{b \in B}\sum_{j \in J_b} \rho^{pen} z_{b,j}^{q} + \delta T_s \left[ \sum_{t \in T}\sum_{i \in I}\sum_{b \in B}\sum_{j \in J_b} \left( \rho_t^{elect} + \rho_t^{em} \right) P_{b,j,i,t}^{q} \right] \tag{24}$$
$$+ \delta \sum_{i \in I} \rho^{DC} P_i^{d,q} \le \sigma \qquad \forall q \le n$$

$$S_{b,j,i}^{dep,q} - d_{b,j}\left( e^{base} + e^{batt} Q_b \right) \ge \vartheta^{min} Q_b \qquad \forall b \in B, \forall j \in J_b \backslash 1, i \in I, \forall q \le n \tag{25}$$

$$S_{b,j,i}^{dep,q} \le \vartheta^{max} Q_b \qquad \forall b \in B, \forall j \in J_b \backslash 1, i \in I, \forall q \le n \tag{26}$$

$$S_{b,j+1,i'}^{dep,q} \le S_{b,j,i}^{dep,q} - d_{b,j}\left( e^{base} + e^{batt} Q_b \right) + \sum_{t \in R_{b,j,i'}} \eta^{ch} T_s P_{b,j,i',t}^{q}$$
$$+ Q^{max} \sum_{j' \le j} z_{b,j'}^{q} \ \forall b \in B, \forall j,j' \in J_b \backslash 1, i\&i' \in I, \quad \forall q \le n \tag{27}$$

$$P_{b,j,i,t}^{q} \le P_{po}^{max} y_{b,j,i,t}^{q} \forall b \in B, \forall j \in J_b, i \in I, \forall t \in R_{b,j,i}, \qquad \forall q \le n \tag{28}$$

$$P_{b,j,i,t}^{q} \le \lambda_1 Q_b \qquad \forall b \in B, \forall j \in J_b, i \in I, \forall t \in R_{b,j,i}, \forall q \le n \tag{29}$$

$$\alpha_{b,j,i,t}^{q} \ge y_{b,j,i,t}^{q} - y_{b,j,i,t+1}^{q} \qquad \forall b \in B, \forall j \in J_b, i \in I, \forall t \in R_{b,j,i}, \forall q \le n \tag{30}$$

$$\gamma_{b,j,i,t}^{q} \ge y_{b,j,i,t}^{q} - y_{b,j,i,t-1}^{q} \qquad \forall b \in B, \forall j \in J_b, i \in I, \forall t \in R_{b,j,i}, \forall q \le n \tag{31}$$

$$\sum_{t \in R_{b,j,i}} \alpha_{b,j,i,t}^{q} = \sum_{t \in R_{b,j,i}} \gamma_{b,j,i,t}^{q} \le 1 \qquad \forall b \in B, \forall j \in J_b, i \in I, \forall t \in R_{b,j,i}, \forall q \le n \tag{32}$$

$$\sum_{b \in B} y_{b,j,i,t}^{q} \le N_i^{po} \qquad \forall i \in I, j, \in J_b, \forall t \in T, \forall q \le n \tag{33}$$

$$\sum_{b \in B} P^q_{b,j,i,t} \leq P^{st}_i \left(1 - \hat{\xi}^q_i\right) \qquad \forall i \in I, j \in J_b, \forall t \in T, \forall q \leq n \qquad (34)$$

$$\sum_{b \in B} P^q_{b,j,i,t} \leq P^{st}_i \qquad \forall i \in I, j \in J_b, \forall t \in T, \forall q \leq n \qquad (35)$$

$$\sum_{b \in B} P^q_{b,j,i,t} \leq P^{\max,st}_{i,t} \qquad \forall i \in I, \forall t \in T, \forall q \leq n \qquad (36)$$

$$\sum_{i \in I}\sum_{b \in B} P^q_{b,j,i,t} \leq P^{\max,net}_t \qquad \forall t \in T, \forall q \leq n \qquad (37)$$

$$P^{avg,q}_{i,\omega} = \frac{\sum_{t \in T_\omega}\sum_{b \in B} P^q_{b,j,i,t}}{|T_\omega|} \qquad \forall i \in I, \forall \omega \in \Omega, \forall q \leq n \qquad (38)$$

$$P^{d,q}_i \geq P^{avg,q}_{i,\omega} \qquad \forall i \in I, \forall \omega \in \Omega, \forall q \leq n \qquad (39)$$

$$
\begin{aligned}
&x_i \in \{0,1\}, P^{st}_i \in A^{st}, N^{po}_i \in Z_{0+} && \forall i \in I\\
&Q_b \in A^{batt} && \forall b \in B\\
&z^q_{b,j} \in \{0,1\} && \forall b \in B, \forall j \in J_b, \forall q \leq n\\
&y^q_{b,j,i,t} \in \{0,1\}, \alpha^q_{b,j,i,t} \in \{0,1\}, \gamma^q_{b,j,i,t} \in \{0,1\}, P^q_{b,j,i,t} \geq 0 && \forall b \in B, \forall j \in J_b, i \in I, \forall t \in R_{b,j,i}, \forall q \leq n\\
&P^{d,q}_i \geq 0 && \forall i \in I, \forall q \leq n\\
&P^{avg,q}_{i,\omega} \geq 0 && \forall i \in I, \forall \omega \in \Omega, \forall q \leq n\\
&S^{dep,q}_{b,j,i} \geq 0 && \forall b \in B, \forall j \in J_b, i \in I, \forall q \leq n
\end{aligned}
$$
$$(40)$$

[SP]:

$$\max_{\xi \in \Xi} \min_{z,y,\alpha,\gamma,P,P^d,P^{avg},S^{dep}} \left[\sum_{b \in B}\sum_{j \in J_b} \rho^{pen} z_{b,j} + \delta T_s \left(\sum_{t \in T}\sum_{i \in I}\sum_{b \in B}\sum_{j \in J_b} \left(\rho^{elect}_t + \rho^{em}_t\right) P_{b,j,i,t}\right) + \delta\sum_{i \in I} \rho^{DC} P^d_i \right]$$
$$(41)$$

Subject to: (6), (8–10), (14–17), (19),

$$S^{dep}_{b,j,i} - d_{b,j}\left(e^{base} + e^{batt}\hat{Q}_b\right) \geq \vartheta^{\min}\hat{Q}_b \qquad \forall b \in B, \forall j \in J_b 1, i \in I \qquad (42)$$

$$S^{dep}_{b,j,i} \leq \vartheta^{\max}\hat{Q}_b \forall b \in B, \qquad \forall j \in J_b 1, i \in I \qquad (43)$$

$$
\begin{aligned}
S^{dep}_{b,j+1,i'} \leq S^{dep}_{b,j,i} &- d_{b,j}\left(e^{base} + e^{batt}\hat{Q}_b\right) + \sum_{t \in R_{b,j,i'}} \eta^{ch} T_s P_{b,j,i',t}\\
&+ Q^{\max}\sum_{j' \leq j} z_{b,j'} \qquad \forall b \in B, \forall j,j' \in J_b 1, i \& i' \in I
\end{aligned}
$$
$$(44)$$

$$P_{b,j,i,t} \leq \lambda_1 \hat{Q}_b \qquad \forall b \in B, \forall j \in J_b, i \in I, \forall t \in R_{b,j,i} \qquad (45)$$

$$N^{po}_i \leq N^{\max}_i \hat{x}_i \qquad \forall i \in I \qquad (46)$$

$$\sum_{b \in B} y_{b,j,i,t} \leq \hat{N}^{po}_i \qquad \forall i \in I, j, \in J_b, \forall t \in T \qquad (47)$$

$$\sum_{b \in B} P_{b,j,i,t} \leq \hat{P}^{st}_i \qquad \forall i \in I, j \in J_b, \forall t \in T \qquad (48)$$

$$\sum_{b \in B} P_{b,j,i,t} \leq \hat{P}^{st}_i (1 - \xi_i) \qquad \forall i \in I, j \in J_b, \forall t \in T, \xi \in \Xi \qquad (49)$$

$$
\begin{aligned}
&z_{b,j} \in \{0,1\} && \forall b \in B, \forall j \in J_b\\
&y_{b,j,i,t} \in \{0,1\}, \alpha_{b,j,i,t} \in \{0,1\}, \gamma_{b,j,i,t} \in \{0,1\}, P_{b,j,i,t} \geq 0 && \forall b \in B, \forall j \in J_b, i \in I, \forall t \in R_{b,j,i}\\
&S^{dep}_{b,j,i} \geq 0 && \forall b \in B, \forall j \in J_b, i \in I\\
&P^d_i \geq 0 && \forall i \in I\\
&P^{avg}_{i,\omega} \geq 0 && \forall i \in I, \forall \omega \in \Omega\\
&\xi_i \in \{0,1\} && \forall i \in I
\end{aligned}
$$
$$(50)$$

First, the MP is solved to obtain the optimal first-stage variables representing the optimal infrastructure configuration, such as the allocated charging stations $\hat{x}$, charger-rated power in each station $\hat{P}^{st}$, number of charger poles $\hat{N}^{po}$, and the BEB fleet battery capacities $\hat{Q}$ considering the charging station failure scenarios $q \leq n$ obtained from the SP solutions. Where $q$ is the failure scenario index, and $n$ is the number of scenarios. In the first iteration, the MP is solved with the initial charging station failure scenario ($n = 0$). Then, the output MP solution ($\hat{x}, \hat{P}^{st}, \hat{N}^{po}, \hat{Q}$) are provided to the SP to estimate the worst-case scenario of charging station failure ($\hat{\xi}$) that has the highest operational cost and failed buses (minimum level of service). In the next iteration, the resulting worst-case scenario $q$ ($\hat{\xi}$) is added to the MP (variables $(z^q, y^q, \alpha^q, \gamma^q, P^q, P^{d,q}, P^{avg,q}, S^{dep,q})$ and Constraints (24–40)) to improve the first-stage solution (more robust). The new solution of the MP will satisfy all the added charging station failure scenarios ($q \leq n$) and provide robust BEB infrastructure decisions against these failure scenarios.

As such, in each iteration, the sub-problem is mainly solved to obtain the worst-case charging station failure $\xi \in \Xi$ scenario under the BEB system configuration estimated using the MP to add this scenario in the next iteration. The SP takes the form of max − min.

The outer-level C&CG algorithm iterates until reaching a robust BEB system configuration against all the charging station failure scenarios that increase the second-stage costs. The MP's objective function value is the solution LB. In addition, the summation of the objective function of the SP and the first-stage infrastructure cost denotes the solution UB. The gap between the UB and LB decreases during iterations and the algorithm terminates after reaching a predefined relative optimality gap ($\varepsilon$). It is worth noting that the relative complete recourse property is satisfied in the proposed model. Therefore, there is no need to add the feasibility cuts during the iterations. Moreover, the algorithm guarantees convergence in finite iterations as the extreme points of the feasibility region of the uncertainty variable are finite. Finally, the outer-level C&CG solution procedure framework is summarized in Algorithm 1 in Supplementary Discussion 7.

**Inner-level C&CG solution algorithm.** The inner-level C&CG solution algorithm is utilized to solve the sub-problem (SP) presented in Eqs. (41–50) that takes the form of max − min and supplies the MP with the worst-case charging stations failure scenarios ($\hat{\xi}$). The bi-level SP cannot be directly reduced into a monolithic optimization problem using the Karush-Kuhn-Tucker (KKT) conditions or the strong duality property because of the integer recourse variables in the second level of the SP ($z, y, \alpha, \gamma$). The main idea of the NC&CG method developed in ref. 49 is to solve the bi-level SP by its equivalent tri-level problem. The tri-level problem will take the form of max − min − min that is similar to a two-stage RO model after separating the discrete variables ($z, y, \alpha, \gamma$) from the continuous variables ($P, P^d, P^{avg}, S^{dep}$) in the second-stage problem as presented in Eq. (51). In this case, the recourse variables in the third level are continuous. Therefore, the standard C&CG solution framework could solve the SP. This is called the inner-level C&CG solution

algorithm.

$$\max_{\xi \in \Xi} \min_{z,y,\alpha,\gamma} \sum_{b \in B} \sum_{j \in J_b} \rho^{pen} z_{b,j} + \min_{P,P^d,P^{avg},S^{dep}} \delta T_s \left( \sum_{t \in T} \sum_{i \in I} \sum_{b \in B} \sum_{j \in J_b} \left( \rho_t^{elect} + \rho_t^{em} \right) P_{b,j,i,t} \right) + \delta \sum_{i \in I} \rho^{DC} P_i^d$$
(51)

Subject to: (6), (8–10), (14,17), (19), and (42–50)

After converting the SP to a tri-level model in Eq. (51), the procedure of the C&CG is utilized. In other words, if all the points (scenarios) of the feasibility region of the discrete variables $z,y,\alpha,\gamma$ are included in the problem, the tri-level problem could be converted to an equivalent model presented in Eqs. (52–64), where $r$ is the scenario index and $R$ is the total number of feasible scenarios. Therefore, the SP solution algorithm comprises solving two optimization problems iteratively: the inner master problem (IMP) to find the failure scenario ($\xi$) and the inner sub-problem (ISP) to provide the IMP with the discrete variables $z^r,y^r,\alpha^r,\gamma^r$ scenarios.

$$\max_{\xi \in \Xi, \theta \geq 0, P^r, P^{d,r}, P^{avg,r}, S^{dep,r}} \theta$$
(52)

Subject to:

$$\theta \leq \sum_{b \in B} \sum_{j \in J_b} \rho^{pen} z_{b,j}^r$$

$$+ \min_{P^r, P^{d,r}, P^{avg,r}, S^{dep,r}} \left\{ \delta T_s \left( \sum_{t \in T} \sum_{i \in I} \sum_{b \in B} \sum_{j \in J_b} \left( \rho_t^{elect} + \rho_t^{em} \right) P_{b,j,i,t}^r \right) + \delta \sum_{i \in I} \rho^{DC} P_i^{d,r} : \atop s.t.(54-64) \right\} \forall r \in R$$
(53)

$$S_{b,j,i}^{dep,r} - d_{b,j} \left( e^{base} + e^{batt} \hat{Q}_b \right) \geq \vartheta^{min} \hat{Q}_b \qquad \forall b \in B, \forall j \in J_b \backslash 1, i \in I, \forall r \leq R$$
(54)

$$S_{b,j,i}^{dep,r} \leq \vartheta^{max} \hat{Q}_b \qquad \forall b \in B, \forall j \in J_b \backslash 1, i \in I, \forall r \leq R$$
(55)

$$S_{b,j+1,i'}^{dep,r} \leq S_{b,j,i}^{dep,r} - d_{b,j} \left( e^{base} + e^{batt} \hat{Q}_b \right) + \sum_{t \in R_{b,j,i'}} \eta^{ch} T_s P_{b,j,i',t}^r$$

$$+ Q^{max} \sum_{j' \leq j} z_{b,j'}^r \qquad \forall b \in B, \forall j,j' \in J_b \backslash 1, i \& i' \in I, \forall r \leq R$$
(56)

$$P_{b,j,i,t}^r \leq P_{po}^{max} y_{b,j,i,t}^r \qquad \forall b \in B, \forall j \in J_b, i \in I, \forall t \in R_{b,j,i}, \forall r \leq R$$
(57)

$$P_{b,j,i,t}^r \leq \lambda_1 \hat{Q}_b \qquad \forall b \in B, \forall j \in J_b, i \in I, \forall t \in R_{b,j,i}, \forall r \leq R$$
(58)

$$\sum_{b \in B} P_{b,j,i,t}^r \leq \hat{P}_i^{st} \qquad \forall i \in I, j \in J_b, t \in T, \forall r \leq R$$
(59)

$$\sum_{b \in B} P_{b,j,i,t}^r \leq P_{i,t}^{max,st} \qquad \forall i \in I, \forall t \in T, \forall r \leq R$$
(60)

$$\sum_{i \in I} \sum_{b \in B} P_{b,j,i,t}^r \leq P_t^{max,net} \qquad \forall t \in T, \forall r \leq R$$
(61)

$$P_{i,\omega}^{avg,r} = \frac{\sum_{t \in T_\omega} \sum_{b \in B} P_{b,j,i,t}^r}{|T_\omega|} \qquad \forall i \in I, \forall \omega \in \Omega, \forall r \leq R$$
(62)

$$P_i^{d,r} \geq P_{i,\omega}^{avg,r} \qquad \forall i \in I, \forall \omega \in \Omega, \forall r \leq R$$
(63)

$$P_{b,j,i,t}^r \geq 0 \qquad \forall b \in B, \forall j \in J_b, i \in I, \forall t \in R_{b,j,i}, \forall r \leq R$$

$$P_i^{d,r} \geq 0 \qquad \forall i \in I, \forall r \leq R$$

$$P_{i,\omega}^{avg,r} \geq 0 \qquad \forall i \in I, \forall \omega \in \Omega, \forall r \leq R$$

$$S_{b,j,i}^{dep,r} \geq 0 \qquad \forall b \in B, \forall j \in J_b, i \in I, \forall r \leq R$$
(64)

A matrix formulation of the equivalent model in Eqs. (52–64) is presented in Eqs. (65–66) to simplify the following steps. Generally, to convert the optimization problem in Eqs. (65–66) to a monolithic optimization problem, classical KKT conditions or the strong duality property could be utilized[50]. However, the extended relative complete recourse is not guaranteed in the proposed model case. Therefore, only strong duality could be used[49].

$$\max_{\xi \in \Xi, \theta \geq 0, P^r \geq 0, P^{d,r} \geq 0, P^{avg,r} \geq 0, S^{dep,r} \geq 0} \theta$$
(65)

Subject to:

$$\theta \leq hZ^r + \min \left\{ gY^r : AY^r \geq f + D\xi, Y^r = \left[ P^r, P^{d,r}, P^{avg,r}, S^{dep,r} \right] \right\} \forall r \in R$$
(66)

Let $\pi^r$ denotes the dual variable to the continuous variable $Y^r = [P^r, P^{d,r}, P^{avg,r}, S^{dep,r}]$ of the minimization problem in the $r^{th}$ constraint of (66). Then, the equivalent monolithic formulation of the problem in Eqs. (65 and 66) is presented in Eqs. (67 and 68). Even though this new formulation is single-level programming, the problem becomes non-linear programming because of the multiplication of $\xi$ and the dual variable $\pi^r$ in Eq. (68). As the failure random variable $\xi \in \Xi$ is binary, Eq. (68) could be linearized by the big − M method. Let $V^r$ are auxiliary variables used in the linearization process.

$$\max_{\xi \in \Xi, \theta \geq 0, \pi^r \geq 0} \theta$$
(67)

Subject to:

$$\theta \leq hZ^r + (f + D\xi)^t \pi^r \qquad \forall r \leq R$$
(68)

$$A^t \pi^r \leq g^t \qquad \forall r \leq R$$
(69)

As such, the final formulation of the inner master problem (IMP) is presented in Eqs. (70–74). This last formulation is linear programming and could be solved effectively using an over-the-shelf solver (e.g., CPLEX, GUROBI). Moreover, it is not required to include all the points in the feasibility region of the discrete variables $z,y,\alpha,\gamma$ in the IMP solution. These scenarios will be added one by one in an iterative process. Therefore, $R$ is replaced by $s$, which represents the iteration number. In each iteration, the values of $z^r,y^r,\alpha^r,\gamma^r$ in the IMP will be obtained by the solution of the inner sub-problem (ISP).

[IMP]

$$\max_{\xi \in \Xi, \theta \geq 0, \pi^r \geq 0, V^r \geq 0} \theta$$
(70)

Subject to:

$$\theta \leq hZ^r + f^t \pi^r + \sum_l V_l^r \qquad \forall r \leq s$$
(71)

$$A^t \pi^r \leq g^t \qquad \forall r \leq s$$
(72)

$$V^r \leq M\xi \qquad \forall r \leq s$$
(73)

$$V^r \leq D^t \pi^r \qquad \forall r \leq s$$
(74)

During the solution of the IMP, the resulting charging station failure variable $\xi$ will be provided to the ISP. Then, the ISP is solved to estimate the minimum operational cost under this failure scenario. The values of the discrete variables will be taken and included in the next iteration of the IMP by adding new variables ($P^r, P^{d,r}, P^{avg,r}, S^{dep,r}$) and Constraints (71–74). The IMP and the ISP's objective function values are considered the IUB and ILB, respectively. Iteratively, the gap will be reduced, and the inner-level C&CG algorithm will converge to the worst-case charging station failure scenario $\hat{\xi}$ with respect to the BEB system infrastructure configuration provided by the MP. Finally, the inner-level C&CG framework summary is presented in Algorithm 2 in the Supplementary Discussion 7.

[ISP]

$$\min_{z,y,\alpha,\gamma,P,P^d,P^{avg},S^{dep}} \left[ \sum_{b \in B} \sum_{j \in J_b} \rho^{pen} z_{b,j} + \delta T_s \left( \sum_{t \in T} \sum_{i \in I} \sum_{b \in B} \sum_{j \in J_b} \left( \rho_t^{elect} + \rho_t^{em} \right) P_{b,j,i,t} \right) + \delta \sum_{i \in I} \rho^{DC} P_i^d \right] \tag{75}$$

Subject to: (6), (8–10), (14–17), (42–48),

$$\sum_{b \in B} P_{b,j,i,t} \leq \hat{P}_i^{st} \left( 1 - \hat{\xi}_i \right) \qquad \forall i \in I, j \in J_b, \forall t \in T, \xi \in \Xi \tag{76}$$

$$
\begin{aligned}
&z_{b,j} \in \{0,1\} && \forall b \in B, \forall j \in J_b \\
&y_{b,j,i,t} \in \{0,1\}, \alpha_{b,j,i,t} \in \{0,1\}, \gamma_{b,j,i,t} \in \{0,1\}, P_{b,j,i,t} \geq 0 && \forall b \in B, \forall j \in J_b, i \in I, \forall t \in R_{b,j,i} \\
&P_i^d \geq 0 && \forall i \in I \\
&P_{i,\omega}^{avg} \geq 0 && \forall i \in I, \forall \omega \in \Omega \\
&S_{b,j,i}^{dep} \geq 0 && \forall b \in B, \forall j \in J_b, i \in I
\end{aligned} \tag{77}
$$

## Reporting summary

Further information on research design is available in the Nature Portfolio Reporting Summary linked to this article.

## Data availability

The Oakville Transit General Transit Feed Specification (GTFS) data used in this study is publicly available at (https://transitfeeds.com/p/oakville-transit/615). Source Data file has been deposited in Figshare under the accession code https://doi.org/10.6084/m9.figshare.24578710[51].

## Code availability

The mathematical formulation and algorithms detailed in this work are scripted in MATLAB (https://www.mathworks.com/products/matlab.html), and the obtained MILPs are solved using Gurobi solver (https://www.gurobi.com/). Custom codes used in this study are available from https://doi.org/10.5281/zenodo.10114326[52].

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

## Acknowledgements

The authors would like to acknowledge support from the Natural Sciences and Engineering Research Council of Canada (NSERC) Grant No: RGPIN-2018-05994, awarded to M.M. Furthermore, this research is partially supported by ASPIRE, the technology program management pillar of Abu Dhabi's Advanced Technology Research Council (ATRC), via the ASPIRE Award for Research Excellence initiative, awarded to E.E.

## Author contributions

A.F. and M.M. conceptualized the idea; A.F. and M.M. synthesized the literature review; A.F. and M.M. structured the data analysis procedures; A.F. and M.M. manipulated the case study data; A.F. completed the coding, software, and mathematical formulation; A.F. executed the experimental simulation work; A.F., and M.M. interpreted the results; A.F., M.M., H.F., and E.E. concluded the work; A.F. and M.M. wrote the manuscript. A.F., M.M., H.F., and E.E. revised the manuscript; M.M. supervised the research; M.M. and E.E. administrated the project, provided resources, and secured the funds.

## Competing interests

The authors declare no competing interests.
