## [Peer review file · Nature Communications]

REVIEWER COMMENTS

Reviewer #1 (Remarks to the Author):

Summary:

To address potential charging station failures, the authors have developed a two-stage robust optimization model and a complicated solution algorithm, referred to as column-and-constraint-generation. This model aims to determine the following: 1) whether a charging station should be constructed for each candidate charging station using a binary decision; 2) the optimal number of charger poles at each constructed charging station; 3) the discrete selection of charger unit-rated power for each constructed charging station; 4) the discrete selection of battery size for each bus within the fleet; 5) the continuous charging power for each bus at a charging station during specific time slots, ensuring each charging procedure a uninterrupted charging process. In this hypothetical scenario, the failure of a charging station is simplified to a binary formulation, indicating whether it is working or not. Furthermore, the maximum allowable number of charging station failures for each bus line is constrained by a predefined budget set.

The authors have successfully formulated and addressed a well-defined mathematical problem encompassing strategic, tactical, and operational decision-making levels for the bus company. However, in order to reinforce the validity of the current charging station failure/closure settings while assuming the normal functioning of other bus system components, the authors should provide comprehensive supplementary data, materials, and real-world examples for thorough discussion.

Some other detailed questions and comments are below:

1. How were those candidate charging stations predetermined?
2. Why was the demand charge not included in the objective function? Why was the emission cost taken into account?
3. The set of timeslots, R_{bj} , was not defined in Table 8.
4. T should be the set of timeslots within a workday?
5. It seems unusual that the timetable remains unchanged in the presence of disruptions.
6. Although the operational timetable is assumed to be fulfilled, it is not clear why backup buses are not utilized. If there are no backup buses, the reason behind this assumption should be explained. Additionally, it would be helpful to understand the impact of the number of buses on the system.
7. It would be beneficial if the authors could provide further elaboration on the settings related to the charging process and energy consumption. For instance, providing more details on the adoption of a constant energy consumption rate and the utilization of a battery-size related energy consumption rate for measuring the energy consumed. Additionally, it would be helpful to have an explanation of the charging limit factors λ_1 and λ_2 and how they are utilized to restrict the charging power.
8. Given that P_{st_i} and Q_b are discrete variables, it may be necessary to introduce auxiliary binary variables to express the corresponding constraints. It would be valuable for the authors to clarify why these variables are not simply treated as continuous.
9. There seems to be an issue with constraints (14) and (15) as their left-hand sides appear to be identical.
10. Regarding the input parameters:

- 1) Could the authors provide more details on how they determined the sets A_{st} and A_{batt} ?
- 2) Was the timeslot duration specified as 2 hours or minutes?
- 3) The parameter δ was set as 365. Does this mean charging station failures are assumed to last for 365 days? It seems unusual for charging station failures to persist for such a long duration. The duration of the disruptions is expected to have a significant impact on the numerical results.
- 4) How was the penalty cost for a failed bus determined? Was a suitably large value chosen?
- 5) A few parameters were missing and not specified in Table 11.

11. The authors may consider conducting additional sensitivity analysis on the adopted parameters to understand their impact on the results. This analysis would provide valuable insights and could be accompanied by thorough discussions on the implications of parameter variations.

12. In regards to the experiment of the solution algorithm, it would be beneficial for the authors to discuss why the lower bound remains relatively stable throughout the iterations. Providing an analysis or explanation for this phenomenon would enhance the understanding of the algorithm's behavior and shed light on the underlying dynamics of the problem.

Reviewer #2 (Remarks to the Author):

The topic addressed by the paper is indeed highly relevant and the robust optimization approach is well appropriate and original for dealing with disruption uncertainty in the design of BEB transit systems. The paper is well written and the mathematics are correct. I would recommend publication with the minor changes:

-Line 139: It is stated than "the daily operation is reduced by more than 25%". Please precise that is occurs for Stations 1 and 4 only.

-Table 3 and 4 should be given also for robust models with $k = 1$ and 2 . Instead, we have only in Table 7 a short summary of service reduction. The more detailed resulted should be given and discussed.

-Line 219: It is stated than the prices of robustness of 3,17% and 9,16% are marginal. Not sure 10% increase is marginal. This may be acceptable, it really depends on the context. this statement needs to be tempered.

Response Letter

In this response letter, detailed discussions are presented on our response to the comments/questions raised during the review process, along with the required references. In this respect, we present the comments of the reviewers in **blue text**, our responses and comments in black text, and the relevant revised text is shown in **italics red**. Furthermore, all modifications in the revised manuscript are highlighted in yellow.

1. Reviewer # 1

Summary: To address potential charging station failures, the authors have developed a two-stage robust optimization model and a complicated solution algorithm, referred to as column-and-constraint-generation. This model aims to determine the following: 1) whether a charging station should be constructed for each candidate charging station using a binary decision; 2) the optimal number of charger poles at each constructed charging station; 3) the discrete selection of charger unit-rated power for each constructed charging station; 4) the discrete selection of battery size for each bus within the fleet; 5) the continuous charging power for each bus at a charging station during specific time slots, ensuring each charging procedure a uninterrupted charging process. In this hypothetical scenario, the failure of a charging station is simplified to a binary formulation, indicating whether it is working or not. Furthermore, the maximum allowable number of charging station failures for each bus line is constrained by a predefined budget set. The authors have successfully formulated and addressed a well-defined mathematical problem encompassing strategic, tactical, and operational decision-making levels for the bus company.

However, in order to reinforce the validity of the current charging station failure/closure settings while assuming the normal functioning of other bus system components, the authors should provide comprehensive supplementary data, materials, and real-world examples for thorough discussion.

- We would like to thank the reviewer for their comprehensive thoughts and detailed comments. We firmly believe that addressing these comments will substantially enhance the quality of the manuscript.
- Here, we addressed the overall comment (**the authors should provide comprehensive supplementary data, materials, and real-world examples for thorough discussion**). After that, detailed responses to the (**other detailed questions and comments**) are presented.
- We totally agree with the reviewer that providing a thorough discussion of the validity of the charging failure/closure settings supported by real-world examples is indeed required.
- In the proposed model, the disruption is assumed to affect a part of the charging system, specifically, k charging stations (where k is the conservative level), while assuming the normal operation of other system components. For example, an equipment malfunction of one charging station ($r = 1$) will impact only one location, while the remainder of the charging infrastructure network will remain operational.
- Our model presented different K values, and we recommend that selecting the value of k should be decided by service providers, as increasing the conservative level k will increase the total system costs.
- Disruption/closure of k charging stations (part of the charging system) could happen due to several reasons related to the utility grid and/or the charging system component failure. Both disruption scenarios are detailed below with real-world data.
- To better communicate this point and as suggested by the reviewer, we have created a new section (Supplementary Materials I). This includes a detailed discussion and real-world examples of various reasons for the partial disruption of BEBs' charging infrastructure.
- Furthermore, we modified the first section of the paper to highlight this new section as follows:
“However, BEB transit system operation is susceptible to several disruptions, such as power outages and equipment malfunction (real-world examples are detailed in Supplementary Materials I).
- Supplementary Materials I: Charging system disruption:
There are several reasons that could lead to the partial failure of BEB's charging system, while the remaining charging network is fully functional. These reasons could be summarized into two categories: utility grid disruption and charging component disruption.

First, the disruption of the utility grid could affect a part of the BEB charging system (specific area) for various reasons ranging from technical issues to external factors. Some of these reasons and real-world examples are as follows:

- 1- Equipment failure of the power grid distribution infrastructure, such as transformers, substations, and circuit breakers, could affect a specific region in the city. For example, an equipment failure in the Hydro Ottawa Richmond Road substation on March 7, 2020, in Ottawa, Canada, affected nearly 12,000 customers for around 20 hours in 5 regions (<https://www.lifebynumbers.ca/history/ottawa-blackouts/>).
- 2- Scheduled maintenance or repairs to the utility grid infrastructure could require shutting down the power in a specific area. For example, the BC Hydro Power Smart, British Columbia, Canada, provides a plan for the number of scheduled electricity outages, their region, duration, and number of customers affected (<https://www.bchydro.com/power-outages/app/outage-list-planned.html#planned>).
- 3- Scheduled load shedding is often used to manage overload and prevent blackouts. This involves shutting down the power to specific regions for a certain period to balance the load. For example, in Chengdu, China, due to extremely high temperatures, 100 electric vehicle charging poles were out of service or limited power during August 2022 (<https://baijiahao.baidu.com/s?id=1741923750151077671&wfr=spider&for=pc>). Similarly, in Egypt, during the summer of 2023, the government mandated a load-shedding program due to the severe heatwave and significant increase in electricity consumption. The following schedule determines power outage regions in each hour (<https://www.egypttoday.com/Article/1/125980/Cabinet-announces-load-shedding-schedule-in-Egyptian-governorates>).
- 4- Many weather-related factors, such as heavy winds, lightning strikes, ice accumulation, or flooding, can damage power lines and equipment, resulting in specific area outages. For example, in Toronto, Canada, parts of the downtown area were left without power due to strong winds in Southern Ontario in December 2022 (<https://www.cp24.com/news/strong-winds-cause-power-outages-across-the-gta-1.6179614>).
- 5- Other factors, such as power line theft, animal interference, human error, and equipment malfunction (faulty relay or sensor), could lead to a specific area of electricity outage. For more details, the distribution of 399 electricity outages in Canada is detailed in (<https://www.eaton.com/content/dam/eaton/products/backup-power-ups-surge-it-power-distribution/backup-power-ups/blackout-tracker-/eaton-blackout-tracker-annual-report-canada-2017.pdf>) and a comprehensive data on major power outage events in the continental U.S is presented in ¹.

In general, power outages are rising due to climate change, increased energy demand, and an aging electrical grid. The USA outage data for 2,447 counties from 2018-2020 shows a total of 17,484 (+8 hours) outages, most of which co-occur with weather-related issues (62.1%) ².

Secondly, scheduled maintenance is required for various components, which could lead to a partial charging service disruption. For example, according to the ABB E-Bus Charger User and Operation Manual for EU products, the charger must be inspected and serviced yearly, including cabinet inspection, components testing, and emergency stop inspection ([https://library.e.abb.com/public/13b2e0d5cfa475188483212fce4d708/HVC%20C%20107-160%20kW%20User%20Manual%20\(v2%20cabinet%20and%20Gen2%20Single%20Depot%20Box\).pdf](https://library.e.abb.com/public/13b2e0d5cfa475188483212fce4d708/HVC%20C%20107-160%20kW%20User%20Manual%20(v2%20cabinet%20and%20Gen2%20Single%20Depot%20Box).pdf)). Moreover, the charger must be checked before reuse in some cases, such as lightning struck, damage due to accident or fire, and its location flooding.

The work of Raman et. al., 2022 highlights the effect of flooding on public electric vehicle charging stations. The authors emphasize that flooding could make chargers unavailable (out of service due to water damage) or disrupt access to the charging stations ³.

- Furthermore, and as suggested by the reviewer, we expanded our Supplementary Materials Section to communicate more data with the readers.
- The content of the revised supplementary materials is summarized as follows:

Section ID	Section Title
I	Charging system disruption
II	Solution algorithm convergence
III	Charging station configuration and utilization (Base Model, Robust Models $k=1$, and $k=2$)
IV	Charging demand distribution of Bus ID 3 under disruption scenarios
V	Daily operation costs under disruption scenario (Robust Models)
VI	Service reduction during disruption
VII	Sensitivity analysis
VIII	Case Study two – Guelph Transit
XI	Solution Algorithms
X	Computational performance enhancement

Some other detailed questions and comments are below:

- We are very pleased to read your comments. We believe that addressing your insightful comments will significantly enhance the quality and contributions of the manuscript. For that, we are very thankful.

1.1. How were those candidate charging stations predetermined?

- We agree that the criteria for choosing the candidate charging stations require more clarification in the manuscript.
- The candidate charging stations are selected from transit hubs, terminals, and intermediate stops based on three measures as follows:
 - 1- The weighted degree centrality of each en-route bus stop, terminal, or hub is determined by the number of buses recovering at that specific location. Locations with a higher weighted degree are selected.
 - 2- The start, end, and any other en-route stop with lengthy recovery time ($\geq 2 T_s$) were selected for each route.
 - 3- For each route, the number of candidate charging stations should be at least $k + 1$, where k is the budget of the charging station failure (conservative level). This condition ensures that the resulting solution is flexible to address the charging station failure uncertainty using all the available decisions (charging station location and configuration, onboard battery capacity, and charging schedule).
- The general proposed criteria for selecting the candidate charging stations are added to the Methods Section as follows.

“The selection of candidate charging station locations is based on three measures. First, the weighted degree centrality (the number of buses recovering) of each bus stop, terminal, or hub in the network is estimated, and stations with a higher weighted degree are prioritized. Second, for each route, the start, end, and any other en-route stop with lengthy recovery time ($\geq 2 T_s$) were considered. Third, the number of en-route candidate charging stations in the routes for each bus should be at least $k + 1$, where k is the budget of the charging station failure. The rationale is to ensure that the resulting solution is flexible to address the failure uncertainty using all the available decisions (charging station location and configuration, onboard battery capacity, and charging schedule).”

- The application of the proposed criteria for selecting the candidate charging stations on Oakville City data (case study) is updated in the Extended Data II Section.

“As mentioned in the Methods Section, selecting the locations of candidate charging stations is based on three measures. First, bus stops, terminals, or hubs with a higher weighted degree of centrality are chosen. The weighted degree centrality of each station in the Oakville network (1,240 locations) is calculated, and stations with a weighted degree higher than 20 are selected (21 locations). In addition, for each route, the start, end, and any other en-route stops with lengthy recovery times were included in the candidate charging station set. This was done while guaranteeing that the count of candidate locations of each route should be higher than $k + 1$. In the case study, the maximum k equals 2. Therefore, each route should include at least three candidate locations of charging stations. Toward this end, a total of 77 en-route locations are identified as candidate charging locations.”

1.2. Why was the demand charge not included in the objective function? Why was the emission cost taken into account?

For demand charge

- Thank you for bringing this valuable point to our attention.
- We acknowledge the importance of including the demand charge in the objective function of the proposed model as a part of the operational costs. Therefore, the entire model, solution algorithm, and results are updated to include the demand charge cost.
- The Contributions Section is modified to include the demand charge as one of the proposed model costs.

“The model estimates the optimal locations of the charging stations, the number of poles and charger-rated power (heterogeneous), and the BEB fleet battery capacities (heterogeneous). The model minimizes the BEB system infrastructure and operational costs considering the electricity Time-of-Use (ToU) tariff and demand charges, the temporal distribution of the GHG emissions, and the utility grid limit constraints.”

- The Methods Section is modified by adding the demand charge in the objective function and related constraints to the model.

$$\min_{x, P^{st}, N^{po}, Q, z, y, \alpha, \gamma, P, P^d, P^{avg}, S^{dep}} \tau \sum_{i \in I} \rho^{st} x_i + \tau \sum_{i \in I} (\rho^{ch} P_i^{st} + \rho^{po} N_i^{po}) + \tau \sum_{b \in B} (\rho^{batt} Q_b + \rho^{bus}) + \sum_{b \in B} \sum_{j \in J_b} \rho^{pen} z_{b,j} + \delta T_s \left[\sum_{t \in T} \sum_{i \in I} \sum_{b \in B} \sum_{j \in J_b} (\rho_t^{elect} + \rho_t^{em}) P_{b,j,i,t} \right] + \delta \sum_{i \in I} \rho^{DC} P_i^d \quad (1)$$

“In Equation (1), the construction cost of charging stations is (ρ^{st}), followed by the cost of charging components (charger units ρ^{ch} and charger poles ρ^{po}). The fleet cost includes batteries ρ^{batt} and buses ρ^{bus} . The penalty cost (ρ^{pen}) is allocated for unsatisfied trips due to failed buses if they exist. The operational costs include electricity time-of-use (ToU) tariffs (ρ_t^{elect}), WTT GHG emissions (ρ_t^{em}), and demand charges (ρ^{DC}). The first two parts of the operational costs are related to the power to charge each bus b in location i after serving segment j at timeslot t ($P_{b,j,i,t}$) and the latter is related to the daily peak power demand (P_i^d).”

“The electricity demand charges are determined by the peak power demand recorded during a billing period, which is measured as the maximum average electric power consumed within a specific time interval (i.e., one hour)⁴. The average power of each demand measurement interval $\omega \in \Omega$ in station i is calculated based on Equation (16), where Ω is the set of daily demand charges measurement intervals and T_ω is the set of timeslots in demand interval ω . In turn, in Equation (17), the maximum daily power demand P_i^d in charging station i is estimated.”

$$P_{i,\omega}^{avg} = \frac{\sum_{t \in T_\omega} \sum_{b \in B} P_{b,j,i,t}}{|T_\omega|} \quad \forall i \in I, \forall \omega \in \Omega \quad (16)$$

$$P_i^d \geq P_{i,\omega}^{avg} \quad \forall i \in I, \forall \omega \in \Omega \quad (17)$$

- The two-stage robust model and the solution algorithm (nested-column-and-constraint solution algorithm) are updated according to the modification of adding the demand charge.
- The modified models (Base Model, Robust Model with $k = 1$, and Robust Model with $k = 2$) are solved, and the Results Section is updated with all the new results of these models after considering the demand charge. For example, the results of the Base Model costs are updated as follows.

“The annual costs of the BEB system are detailed in Table 1, with a total annual cost of \$6,959,381.192. It should be highlighted that the capital expenses are amortized over the 12-year lifespan. The primary contributor is the fleet cost without batteries (76.63%), while operational costs (electricity ToU, demand charge, and GHG emissions) account for 11.58%. In comparison, BEBs' batteries contribute 10.03%, and the charging system (charging stations infrastructure and chargers) accounts for 1.77%.”

Table 1. System annual costs (Base Model)

Parameter	Annual Costs	Percentage (%)
Infrastructure cost	\$ 39,957.065	0.574%

Parameter	Annual Costs	Percentage (%)
Chargers cost	\$ 83,004.143	1.193%
Battery cost	\$ 697,916.731	10.028%
Fleet cost	\$ 5,332,936.243	76.629%
Operational cost	\$ 805,567.010	11.575%
Total annual cost	\$ 6,959,381.192	100.00%

- In addition, the results of the Robust Models ($k = 1&2$) are also updated along with the comparison between the three models. For example, Figure 1 in the Revised Manuscript is modified and includes a comparison of the electricity demand costs between the Robust Models and the Base Model.

Figure 1. Cost comparison between Base Model, Robust Model ($k = 1$), and Robust Model $k = 2$

Why was the emission cost taken into account?

- Including the temporal distribution of the GHG emissions in the proposed model addresses two aspects:
- **First**, the GHG emissions objective function aims to minimize the well-to-tank (WTT) GHG emissions resulting from electricity generation. This is to overcome the fact that WTT GHG emissions vary during the day. Therefore, the resultant charging schedule minimizes the total WTT GHG emissions for the BEB system, which increases BEB's environmental competitiveness. This is supported by the literature such as the work of Rupp, et al. ⁵ and Foda, et al. ⁶.
- **Second**, this term accommodates the new carbon trade policies introduced in Canada and globally to reduce GHG emissions. In Canada, for example, the federal carbon pricing system includes two parts: 1) a regulatory charge on fossil fuels, known as fuel charge, and 2) a performance-based system for industries, known as the Output-Based Pricing System ⁷. Canada's minimum Carbon Pollution Price is 65 \$/tCO₂e in 2023 and will increase by 15\$/tCO₂e each year until 2030. These policies adjust the ToU tariff based on the temporal distribution of the WTT GHG emissions. Therefore, electricity consumers are eagerly trying to avoid on-peak WTT GHG emissions periods. This is addressed in our proposed model by minimizing the WTT GHG emissions.
- The following paragraph is added to the Revised Manuscript to highlight this point.

“It is worth noting that including the temporal well-to-tank (WTT) GHG emissions resulting from electricity generation in the proposed model aims to decrease the overall GHG emissions of the entire system, enhances the environmental competitiveness of the BEB system, and aligns with carbon pollution pricing policies⁷. In Rupp, et al. ⁵, an optimization model is developed to optimize the charging schedule, incorporating the electricity ToU rates and the temporal data of CO₂e emissions per kWh. The research reveals that integrating WTT GHG emissions into the optimization process results in a 14.9% reduction in total system GHG emissions compared to considering only the electricity ToU tariff. Additionally, in Foda, et al. ⁶, when both the ToU of WTT

GHG emissions and electricity tariffs are taken into account, the total system GHG emissions decrease by 13.34%.”

1.3. The set of timeslots, R_{bj} , was not defined in Table 8.

- Nice Catch 😊. Checked and defined.
- All the notations in the proposed model are checked and defined in Table 8.

1.4. T should be the set of timeslots within a workday?

- Noted and modified.

1.5. It seems unusual that the timetable remains unchanged in the presence of disruptions.

- Thank you for the comment.
- Indeed, we agree with the reviewer that maintaining the operational timetable unchanged in the presence of charging station disruption is challenging (Pre-perturbation resilient system design). However, this is our aim. Our fundamental main contribution is developing a resilient BEB system design model that is robust against any k simultaneous charging station failures while maintaining the same level of service (operational timetable).
- In the Contributions Section, we highlighted this point.

“The model guarantees that the optimal BEB system satisfies full operation under any charging station failure scenario while optimizing the BEB system configuration for planning and operation.”

- Transit is mandated to serve customers in a timely manner. Therefore, the timetable is a hard constraint.
- In the Assumptions, we added the following.

“The model satisfies the operational timetable of the transit network⁸.”

- The question of how the buses maintain the exact timetable under charging station disruption is answered by the second-stage (wait-and-see) decision variables. These variables are the actual operation decisions that construct the charging schedule according to the revealed charging station disruption locations. In other words, the obtained BEB system configuration is able to produce an optimal changeable daily charging schedule according to the disruptive locations. Therefore, we can say that the bus timetable is maintained while the bus charging schedule varies according to the disruption. The charging event is happening during the recovery time.
- To clarify this point, the following paragraph is added to the Revised Manuscript.
- As an example, the fleet charging schedule adjustment due to the disruption of Station ID 1 is also added to the Extended Data IV. In addition, Bus ID 3’s energy demand distribution is illustrated in Supplementary Materials III under the disruption of each en-route charging station.

“This resulting BEB system configuration is robust against any single charging station failure. During failure, the model creates a new charging schedule utilizing the remaining charging stations and ensures that all BEBs complete their scheduled trips (An example is illustrated in Extended data IV). For instance, the energy demand of Bus ID 3 is 412.8 kWh drawn from four en-route charging stations ID 4, 7, 19, and 24 in the nominal operation (163.79 kWh, 33.33 kWh, 118.52 kWh, and 97.18 kWh, respectively). In the case of the failure of any of these four charging stations, Bus ID 3 charges the same amount from the remaining three stations.”

- In Extended data IV (Charging schedule adjustment due to disruption (Station ID 1)):

Table 12. Number of charging events (Robust Model, $k = 1$) – No disruption

Hour	Station ID																			
	1	3	4	7	9	10	11	14	16	19	21	24	30	31	33	35	47	72		
05:00	1	2	1	0	0	0	2	0	0	1	1	0	0	1	1	0	0	0	0	
06:00	23	10	13	3	10	7	12	2	1	6	3	8	8	5	8	5	7	5	5	
07:00	19	11	15	7	14	10	9	4	2	13	7	8	13	7	13	5	8	4	4	
08:00	4	7	7	7	14	11	7	4	2	10	6	7	9	6	11	4	8	3	3	
09:00	2	6	4	7	6	6	7	2	1	8	3	6	8	6	11	4	6	4	4	
10:00	0	5	3	4	6	4	6	1	2	6	3	3	9	6	12	4	6	5	5	
11:00	15	11	10	5	6	5	10	2	1	6	3	2	9	6	10	2	6	4	4	
12:00	18	9	7	3	6	4	9	1	2	5	3	4	8	6	12	2	6	4	4	
13:00	12	8	7	5	6	5	9	2	1	6	3	4	8	6	10	2	6	4	4	
14:00	16	9	7	3	6	4	10	1	2	5	3	4	8	6	12	3	6	4	4	

Hour	Station ID																	
	1	3	4	7	9	10	11	14	16	19	21	24	30	31	33	35	47	72
14:00	15	11	7	5	9	5	9	2	2	5	4	4	9	7	11	2	6	4
15:00	17	9	10	6	11	6	7	5	2	8	8	5	12	6	11	2	6	4
16:00	0	11	8	8	16	8	7	3	1	9	8	11	12	7	13	4	6	6
17:00	3	10	9	6	12	7	8	2	2	10	7	9	10	7	11	6	10	5
18:00	18	8	11	5	9	8	7	2	1	9	3	4	8	5	12	5	5	4
19:00	13	5	5	2	4	4	6	2	2	4	3	2	6	4	7	4	3	2
20:00	12	5	7	4	3	4	7	2	2	4	3	2	5	3	6	3	3	2
21:00	13	5	6	2	3	3	8	1	2	3	3	0	6	3	6	2	3	2
22:00	9	5	5	2	2	2	6	2	1	2	1	0	1	2	6	1	2	1
23:00	5	0	0	0	0	0	1	0	0	0	0	0	0	0	1	1	0	1

Table 13. Number of charging events (Robust Model, $k = 1$) – Under disruption of Station ID 1

Hour	Station ID																
	3	4	7	9	10	11	14	16	19	21	24	30	31	33	35	47	72
05:00	2	1	0	0	0	2	0	0	1	1	0	0	1	1	0	0	0
06:00	10	13	3	10	7	12	2	1	6	3	8	8	5	8	5	7	5
07:00	10	14	7	14	10	12	4	2	13	7	8	13	7	12	5	9	4
08:00	6	9	7	14	11	8	4	2	10	6	7	9	6	11	4	8	3
09:00	7	4	7	6	6	7	2	1	9	3	6	7	6	11	4	6	4
10:00	5	3	4	6	4	6	1	2	6	3	3	9	6	12	4	6	5
11:00	9	11	5	6	5	10	2	1	6	3	2	9	6	10	2	5	4
12:00	9	6	3	6	4	9	1	2	5	3	4	9	6	12	2	6	4
13:00	8	8	5	6	5	9	2	1	6	3	4	9	6	10	2	5	4
14:00	9	8	3	6	4	10	1	2	5	3	4	9	6	12	3	5	4
14:00	10	9	5	9	5	9	2	2	5	4	4	9	7	11	2	5	4
15:00	10	11	6	11	6	9	5	2	8	8	5	13	6	11	2	5	4
16:00	11	8	8	16	8	7	3	1	9	8	11	12	7	13	4	6	6
17:00	9	9	6	12	7	9	2	2	10	7	9	10	7	11	6	10	5
18:00	8	13	5	9	8	7	2	1	9	3	4	8	5	11	5	5	4
19:00	5	6	2	4	4	7	2	2	4	3	2	6	5	7	4	3	2
20:00	5	8	4	3	4	7	2	2	4	3	2	5	4	6	3	3	2
21:00	5	6	2	3	3	8	1	2	3	3	0	6	4	7	2	3	2
22:00	5	5	2	2	2	6	2	1	2	1	0	1	2	7	1	2	1
23:00	0	0	0	0	0	1	0	0	0	0	0	0	0	1	1	0	1

- In Supplementary Materials IV (charging demand distribution of Bus ID 3 under disruption):

Table S.4. Energy demand distribution of Bus ID 3 under disruption (Robust Model, $k = 1$)

Disruption Scenarios	Charging demand per station			
	4	7	19	24
No disruption	163.79 kWh	33.33 kWh	118.52 kWh	97.18 kWh
Disruption to Station ID 4	NA	50.00 kWh	234.61 kWh	128.22 kWh
Disruption to Station ID 7	173.92 kWh	NA	125.06 kWh	113.85 kWh
Disruption to Station ID 19	276.59 kWh	16.67 kWh	NA	119.57 kWh
Disruption to Station ID 24	178.54 kWh	55.74 kWh	178.54 kWh	NA

Table S.5. Energy demand distribution of Bus ID 3 under disruption (Robust Model, $k = 2$)

Disruption Scenarios	Charging demand per station				
	4	6	7	19	24
No disruption	66.67 kWh	40.49 kWh	20.13 kWh	180.31 kWh	105.22 kWh
Disruption to Stations ID 4 & 6	NA	NA	33.33 kWh	187.72 kWh	191.77 kWh
Disruption to Stations ID 4 & 7	NA	91.07 kWh	NA	185.24 kWh	136.52 kWh
Disruption to Stations ID 4 & 19	NA	144.32 kWh	45.88 kWh	NA	222.62 kWh
Disruption to Stations ID 4 & 24	NA	138.43 kWh	66.67 kWh	207.72 kWh	NA
Disruption to Stations ID 6 & 7	97.72 kWh	NA	NA	139.61 kWh	175.50 kWh
Disruption to Stations ID 6 & 19	179.60 kWh	NA	33.90 kWh	NA	199.32 kWh
Disruption to Stations ID 6 & 24	196.48 kWh	NA	58.94 kWh	157.40 kWh	NA
Disruption to Stations ID 7 & 19	156.42 kWh	116.18 kWh	NA	NA	140.23 kWh
Disruption to Stations ID 7 & 24	125.00 kWh	125.23 kWh	NA	162.59 kWh	NA

Disruption to Stations ID 19 & 24	229.26 kWh	127.84 kWh	55.72 kWh	NA	NA
------------	------------	-----------	----	----

1.6. Although the operational timetable is assumed to be fulfilled, it is not clear why backup buses are not utilized. If there are no backup buses, the reason behind this assumption should be explained. Additionally, it would be helpful to understand the impact of the number of buses on the system.

- Thank you for the comment and for bringing these valuable points to our attention.
- There are three reasons for not utilizing backup buses to fulfill the trips of the failed buses after charging station disruptions.
- **First** is the high cost of the BEBs. Buying additional buses just to serve the disruption is extremely costly. Even without batteries, bus cost is relatively higher than cost of a charging station. One BEB with a 100 kWh battery could cost around \$600,000, while one charging station with a 250 kW charger with two poles cost approximately \$176,000. In addition, the added charging station could serve multiple buses and prevent their failure. In the Base Model of our study, the fleet cost without batteries is 76.63% of the total system cost, while the charging system accounts for 1.77%.
- Furthermore, the failure of one station in the Base Model leads to up to 30 failed buses, as mentioned in Table 3. However, the required additional total annual system costs to provide a resilient model that is robust against one charging station failure (Robust Model, $k = 1$) is \$227,207.206 (Table 7), which is equal to the cost of four BEBs (this estimation is calculated after annualizing the bus cost).
- In this respect, we are focusing on providing a cost-optimal resilient BEB system configuration..
- To clarify this point, the following assumption is added to the Revised Manuscript.

“No backup buses are utilized to fulfill the trips of the failed buses due to the high added cost.”

- **Second** is the Spare Ratio Policies (reserve fleet ratio). For example, the Circular FTA C 9030.1C of the FTA (Federal Transit Administration, USA) restricts the number of spare buses in any fleet operating more than 50 to a maximum of 20% of the operating fleet size (<https://libraryarchives.metro.net/dpctl/FTA/1998-FTA-Circular-9030.1C-Urbanized-Area-Formula-Section-5307.pdf>).
- In our case, the maximum reserve fleet is 19 BEBs. These 19 BEBs are typically utilized during scheduled maintenance. However, 19 BEBs will not be sufficient to replace the failed buses even for one charging station failure scenario.
- **The third** is related to providing seamless operation. Our aim is to provide a robust BEB system configuration against charging station disruptions. In this resilient infrastructure design, buses will operate smoothly according to the operational timetable with just some changes in the charging schedule in case of charging station disruption. As such, There will be no delays or any inconvenience to the passengers, drivers, or the transit agency due to changing the failed buses by backup buses.
- The following sentence is added to the Revised Manuscript to elaborate on the reasons behind this assumption.

“The rationale of the last assumption is 1) the high cost of the BEB relative to the cost of the charging station, 2) the high number of failed BEBs due to daily charging station disruption (more than the spare ratio), 3) and the need to provide smooth operation to the passengers, drivers, and transit agency even during charging station disruption.”

- Despite these three reasons, we acknowledge the reviewer's point and thank the reviewer for providing us with a valuable research point to be considered in advancing our work.

Additionally, it would be helpful to understand the impact of the number of buses on the system.

- This is a fantastic point that will be thoroughly explored in our future research.
- As per our intuition, the impact of the charging station disruption in the transit network is a function of multiple factors, such as the spatial configuration of the network (network type and graph density), number and specification of the buses, location of the charging stations, number and specifications of the charging stations, the charging schedule, and the operational timetable.
- In addition, our proposed model aims to provide the transit agency with a resilient BEB system planning and operation of the already established transit network to promote transit network electrification. Therefore, our proposed model maintains the exact fossil-fuelled network timetable and the number of buses.

- As such, we cannot estimate the true impact of the number of buses. However, we apply our proposed model to another transit network with a different network structure, operational timetable, and number of buses. We solve the same proposed model using Guelph City, Ontario, Canada, transit network data.
- Overall, comparing the results of the two case studies indicates that multiple factors impact the resiliency of the system. These include, the spatial configuration of the network (network type and graph density), number and specification of the buses, location of the charging stations, number and specifications of the charging stations, charging schedule, and operational timetable.
- The network description and results are reported in Supplementary Materials VIII, and the following sentences are added to the Revised Manuscript.

“The proposed model is applied to another transit network (Guelph City, Ontario, Canada) with different network structure, operational timetable, and number of buses. The detailed results are presented in Supplementary Materials VIII.”

- Supplementary Materials VIII – Guelph Transit network

“The proposed BEB system configuration model is applied to a multiple hub transit network of Guelph City, Ontario, Canada. The network comprises 23 routes served by 55 buses and 506 bus stops/terminals. The Guelph network has 18 transfer stations and terminals, serving multiple routes and long recovery times. These locations are selected as the candidate charging stations, illustrated in Supplementary Figure 5. More details about the network timetable are presented in the work of Foda, et al. ⁶. The proposed model uses the same parameters shown in Table 11.

The solution algorithm converges in four iterations after adding nine failure scenarios. The results of the system configurations of the Base Model and the Robust Model with $k = 1$ are presented in Supplementary Table 14. The optimal BEB system under nominal operation requires ten heterogeneous charging stations equipped with 14 poles. The main station is Station ID 4, which comprises a 1000 kW charger unit with four poles. The disruption of this station leads to 20 failed buses (36% of the total number of buses). In comparison, the resilient model that could fulfill all the operational trips under one station disruption (Robust Model, $k = 1$) requires 17 charging stations equipped with 23 poles.

In Supplementary Table 15, the distribution of the annual system costs of the two models is illustrated. In this network, the price of the robustness (PoR) of the robust BEB system configurations with $k = 1$ is 4.69%. This will prevent the failure of up to 20 BEBs in case of one station failure.

Comparing the results of Oakville and Guelph transit networks emphasizes that the impact of the charging station disruption in the transit network and the price of robustness are functions of multiple factors, such as the spatial configuration of the network (network type and graph density), number and specification of the buses, location of the charging stations, number and specifications of the charging stations, the charging schedule, and the operational timetable.”

Supplementary Figure 5. Candidate charging stations of the Guelph network

Supplementary Table 14. Results of BEB system configuration (Base and Robust Model)

Models	Total annual system cost (\$/year)	Number of buses (#) × Battery size (kWh)	Number of charging stations	Power of charger units (kW)	Number of poles (#)
Base Model	\$ 4,763,136.459	51×100 3×200 1×600	10	8×250 1×500 1×1000	14
Robust Model $k = 1$	\$ 4,986,729.762	35×100 17×200 2×400 1×600	17	6×250 11×500	23

Supplementary Table 15. System annual costs (Base and Robust Model)

Parameter	Base Model	Robust Model, $k = 1$ from
Infrastructure cost	\$ 79,914.130	\$ 135,854.020
Chargers cost	\$ 131,272.277	\$ 255,618.662
Battery cost	\$ 335,639.344	\$ 442,191.517
Fleet cost	\$ 3,223,203.224	\$ 3,223,203.224
Capital costs	\$ 3,770,028.974	\$ 4,056,867.423
Operational costs*	\$ 993,107.485	\$ 929,862.339
Total annual cost	\$ 4,763,136.459	\$ 4,986,729.762

* Operational costs include the electricity ToU, demand charges, and emissions costs and are estimated in the Robust Model based on the scenario of no charging station failures. Please note that the battery capacity of each model is a decision variable, hence the variation of the battery cost across models.

1.7. It would be beneficial if the authors could provide further elaboration on the settings related to the charging process and energy consumption. For instance, providing more details on the adoption of a constant energy consumption rate and the utilization of a battery-size related energy consumption rate for measuring the energy consumed. Additionally, it would be helpful to have an explanation of the charging limit factors λ_1 and λ_2 and how they are utilized to restrict the charging power.

- We agree with the reviewer on the proposed suggestion for further elaboration on these two points.

For the energy consumption estimation:

- In the proposed model, the energy consumption of each bus b during segment j (kW) is estimated by a linear function of the segment length $d_{b,j}$ (km) and the energy consumption rate (kWh/km).
- The energy consumption rate comprises two components: the base component (fixed) and another component related to the bus battery capacity. The existence of the second part is attributed to the fact that the battery capacity of each bus is considered a decision variable in the proposed model. Increasing the bus battery capacity increases the weight of the bus, subsequently leading to a higher energy consumption rate. For example, in Weiss, et al. ⁹, it is found that a battery capacity increase of 10 kWh corresponds to an added mass of 15 kg to the electric bus, resulting in additional energy consumption ranging from 0.7 to 1.0 kWh per 100 km travelled.
- In this respect, the energy consumption rate is formulated as $(e^{base} + e^{batt}Q_b)$, where e^{base} is the energy consumption rate base component without including the battery weight, $e^{batt}Q_b$ is the variable component of the energy consumption rate due to the weight of a battery pack with a capacity Q_b , and e^{batt} denotes the extra energy consumption rate resulting from an increase in the battery size by one unit. This formulation is similar to the work of He, et al. ¹⁰ and Zhou, et al. ¹¹.
- The proposed model is updated, and the following sentences are added to the Methods Section to clarify this point.

“Where $S_{b,j,i}^{dep}$ is the departure battery capacity for bus b to serve segment j from location i and $d_{b,j}(e^{base} + e^{batt}Q_b)$ is the consumed energy in this segment. The energy consumption of bus b during operation in segment j is related to the segment length $d_{b,j}$ and the energy consumption rate. In this work, the energy consumption rate is formulated as $(e^{base} + e^{batt}Q_b)$, where e^{base} is the energy consumption rate for the base component without including the battery weight, $e^{batt}Q_b$ is the variable component of the energy consumption rate due to the weight of a battery pack with capacity Q_b , and e^{batt} denotes the extra energy consumption rate resulting from an increase in the battery size by one unit ^{10,11}. It is worth noting that raising the bus battery capacity

results in an increase in the weight of the battery pack, subsequently leading to a higher bus energy consumption rate⁹.

- Due to the lack of real-world BEB energy consumption data for the Oakville transit, the energy consumption rate parameters e^{base} and e^{batt} are estimated using a linear regression model for data extracted from a BEB simulator. Advanced Vehicle Simulator (ADVISOR) MATLAB/Simulink environment is utilized to extract the BEB energy consumption data for various battery capacities. ADVISOR was first developed by the National Renewable Energy Laboratory (NREL) and used in the literature for simulating the BEB conduct^{12,13}. The Orange County Cycle 1-Hz speed profile is used in the simulation process¹⁴.
- The estimation process of the energy consumption parameters is updated in the Revised Manuscript in the Extended Data II to clarify this point.

“Due to the lack of real-world BEB energy consumption data for the Oakville transit, the energy consumption rate parameters e^{base} and e^{batt} are estimated using a linear regression model for data extracted from a BEB simulator. Advanced Vehicle Simulator (ADVISOR) MATLAB/Simulink environment is utilized to extract the BEB energy consumption data for various battery capacities. ADVISOR was first developed by the National Renewable Energy Laboratory (NREL) and used in the literature for simulating the BEB conduct^{12,13}. The Orange County Cycle 1-Hz speed profile is used in the simulation process¹⁴.”

For the charging process:

- The following paragraph is added to the Revised Manuscript in the Methods Section to further elaborate on the charging process in the proposed model.

“During operation, the recovery time between two consequence trips is exploited to charge BEBs. Specifically, during each recovery time, the decisions to charge and the charging duration are informed through the BEB system configuration optimization. Moreover, each charging event should satisfy two concepts: partial charging, which constrained the charging duration to be less than or equal to the available recovery time, and continuous charging, which restricts the charging to be in a continuous-time interval. In addition, the charging power of each charging event is variable. It is related to several parameters, such as the charger pole-rated power limits, the available power depending on the station’s charger unit-rated power, the number of BEBs charging simultaneously, and the BEB battery capacity.”

Additionally, it would be helpful to have an explanation of the charging limit factors λ_1 and λ_2 and how they are utilized to restrict the charging power.

- The charging limit factors λ_1 and λ_2 are proposed to keep the charging power in each charging event within a range related to the bus battery capacity. In other words, the charging power $P_{b,j,i,t}$ for bus b should be in the range $[\lambda_2 Q_b, \lambda_1 Q_b]$, where Q_b is the battery capacity of the bus b .
- In the previous model solution, λ_2 was set to zero. Therefore, the charging power range was $[0, \lambda_1 Q_b]$.
- Consequently, in the Revised Manuscript, λ_2 limit and the related constraint are removed from the modified models as the charging power $P_{b,j,i,t}$ is defined as a non-negative continuous variable in Equation (18).
- The reason behind using λ_1 limit factor in the proposed model is related to the battery C-rate, which is a measure of the rate at which a battery is charged or discharged relative to its capacity. Multiple studies have indicated that charging under low temperatures and high C-rate can lead to expedited battery capacity degradation¹⁵. Therefore, using high-power chargers (en-route fast charging) to reduce the charging time is a dilemma. In this respect, the λ_1 limit factor is utilized to cap the charging power. Constraint (7) is active mainly when a low bus battery capacity is chosen.
- To clarify this point in the Revised Manuscript, the Equation (7) description is updated as follows.

“In Equation (7), the charging power is lower than a maximum factor (λ_1) multiplied by the battery capacity. This factor is related to the battery C-rate (the rate of charge or discharge relative to the battery capacity). Charging above the C-rate will lead to accelerated battery fading¹⁵.”

$$P_{b,j,i,t} \leq \lambda_1 Q_b \quad \forall b \in B, \forall j \in J_b, i \in I, \forall t \in R_{b,j,i} \quad (7)$$

1.8. Given that P_{st} and Q_b are discrete variables, it may be necessary to introduce auxiliary binary variables to express the corresponding constraints. It would be valuable for the authors to clarify why these variables are not simply treated as continuous.

- Thank you for the comment.
- In the proposed model, the charger-rated power P_i^{st} and the bus battery capacity Q_b are discrete variables. The values of P_i^{st} and Q_b are chosen from pre-determined sets A^{st} and A^{batt} , respectively, necessitating auxiliary binary variables to be introduced in the solution process.
- However, the rationale behind not treating these variables as continuous is related to market availability. The manufacturing companies of the BEBs and bus chargers produce them with specific values/specifications. For example, BYD Company Ltd. provides BEBs with battery capacities such as 215, 313, 391, 446, and 578 kWh (<https://en.byd.com/bus/>). For charger-rated power, as an example, Siemens Company provides charger units with various powers such as 150, 300, 450, and 600 kW (<https://assets.new.siemens.com/siemens/assets/api/uuid:63e3ce6f-36ad-41b1-93d3-9725a25d0460/sicharge-uc-brochure-en.pdf>).
- Therefore, our proposed approach is to make the charger-rated power and the battery capacity chosen from predetermined sets defined using the available values/specifications in the market when the proposed model is utilized. Moreover, fast charger and battery technologies are rapidly evolving.
- The following sentences are updated in the Revised Manuscript.

“Equation (18) imposed the types of variables, such as $x_i, z_{b,j}, y_{b,j,i,t}, \alpha_{b,j,i,t}$ and $\gamma_{b,j,i,t}$ are binary, N_i^{po} is a non-negative integer, and $P_{b,j,i,t}, P_i^d, P_{i,\omega}^{avg}$, and $S_{b,j,i}^{dep}$ are continuous. The variables P_i^{st} and Q_b are selected from predefined finite sets that represent a wide range of the available values/specifications in the market with the recent technology.”

1.9. There seems to be an issue with constraints (14) and (15) as their left-hand sides appear to be identical.

- We agree with the reviewer that the left-hand sides of the two equations are identical. However, the usage of each equation is different.
- The left-hand side in these two equations presents the total charging power of all BEBs charging in location i in timeslot t ($\sum_{b \in B} P_{b,j,i,t}$). However, the right-hand side differs from equation (14) to equation (15).
- In Equation (14), the right-hand side is the charger power in location i (P_i^{st}). Therefore, Constraint (14) is used to ensure that the total charging power of the BEBs in the same location and the same timeslot is limited by the selected charger unit-rated power at this location.
- However, in Equation (15), the right-hand side is the upper bound temporal limit of the total charging power in location i . As such, Constraint (15) is utilized to cap the total charging power in location i during timeslot t by a predefined temporal cap, which is used mainly to limit the power demand in location i during on-peak periods to reduce the impact of the charging system in the utility grid ⁶.
- Please note that the numbers of these equations are changed in the updated model to (13) and (14).

1.10. Regarding the input parameters:

Could the authors provide more details on how they determined the sets A_{st} and A_{batt} ?

- Actually, they are assumed to include a wide range of values representing the entire market.
- In reality, when the model is applied, we recommend using the available values/specifications in the market.
- For the battery capacity, A^{batt} is assumed to be $\{100, 200, \dots, 700\}$ to present a wide range of the available BEBs' battery capacity without restricting to a specific manufacturer. For example,
 - 1- BYD Company Ltd. provides BEBs with battery capacities such as 215, 313, 391, 446, and 578 kWh (<https://en.byd.com/bus/>).
 - 2- New Flyer Company provides BEBs with battery capacities such as 345, 435, 520, and 734 kWh (<https://www.newflyer.com/new-flyer-buses-meet-the-xcelsior-family/>).
 - 3- Proterra Company provides CATALYST 40-foot bus series with battery capacities such as 79, 105, 220, 330, 440, 550, and 660 kWh (<https://www.proterra.com/wp-content/uploads/2016/08/Proterra-Catalyst-Vehicle-Specs.pdf>).
- For the charger power, A^{st} in the modified model is assumed to be $\{250, 500, \dots, 1500\}$ to present a wide range of available chargers. For example,

- 1- Siemens Company has a modular charger with levels 150, 300, 450, and 600 kW (<https://assets.new.siemens.com/siemens/assets/api/uuid:63e3ce6f-36ad-41b1-93d3-9725a25d0460/sicharge-uc-brochure-en.pdf>).
 - 2- ChargePoint Company offers several modular charging systems built to scale. For example, ChargePoint Express Plus 350 kW DC fast charging system with a power block can hold up to 5 power modules (5×350) and deliver power to up to eight power link dispensers (<https://www.chargepoint.com/en-ca/businesses/dc-stations/express-plus>).
 - 3- Proterra Company provides purpose-built charging hardware to support large electric fleets to support up to 48 vehicles with a charging system of 1440 kW (https://www.proterra.com/wp-content/uploads/2023/02/SPEC_CHG-SYS_1440kW_V9_02_20_22.pdf).
- In the Methods section, the following sentence is updated.

"The variables P_i^{st} and Q_b are selected from predefined finite sets that represent a wide range of the available values/specifications in the market with the recent technology."

- In the Extended Data – II Section, the following sentences are added to explain the selection of the values of these sets.

"The values of the sets A^{batt} and A^{st} are assumed to include a wide range of values representing the entire market without any restriction to a specific manufacturer."

Was the timeslot duration specified as 2 hours or minutes?

- Nice Catch. Noted and corrected.
- It is 2 minutes. However, the timeslot duration is modified in Table 11 (Input values of model parameters) to $\frac{2}{60}$ hour as it is defined in Table 8 (Abbreviations and notations) and used in the proposed model in the hour unit.

The parameter δ was set as 365. Does this mean charging station failures are assumed to last for 365 days? It seems unusual for charging station failures to persist for such a long duration. The duration of the disruptions is expected to have a significant impact on the numerical results.

- Thank you for the comment. We agree with the reviewer that the duration of the disruptions impacts the numerical results significantly. However, setting δ to 365 does not mean that the charging station failures are assumed to last for 365 days.
- In the proposed model, the duration of the disruption is assumed to be one daily (for the entire operational day) (Assumption 5) without any restrictions on the number of days for the charging station failure scenario. Each day, the system could have a different charging station failure scenario (including no failure scenario, a scenario ending in one day, or a scenario taking more than one day).
- This entire-day disruption assumption is based on a recent study¹⁶, which shows that current BEB optimization models are robust against disruptions if solved promptly (within one hour). However, daily disruption of one charging station could reduce the frequency of the service by 57%.
- From a modelling perspective, the parameter δ is set to 365 (the number of days in the year), and the annual operational costs in the objective function are estimated by multiplying the daily operational costs by δ . This is to annualize the capital and operational costs. This approach aligns with the two-stage robust optimization, which considers the maximization of the operational costs by getting the daily worst-case scenario of the charging station failure ($\max_{\xi \in \Xi}$) and multiply it by 365 to estimate the worst annual operational costs in the second stage (sub-problem). Toward that end, the proposed two-stage robust model solution provides a resilient BEB system design that is robust against any daily charging station failures within the designed budget k .
- However, this does not mean the charging station failure should be for the entire year. Specifically, every day, the charging station disruption scenario is realized, and the optimal charging schedule is developed using the inner sub-problem in Equations (75-77). In this respect, the operational costs of this day are estimated according to the resulting charging schedule. Then, the total annual operational costs are calculated by adding the daily operational cost of the entire year. As such, each day could have a different charging station failure scenario.
- In Table 6, the reported operational costs are based on the scenario of no charging station failures for the entire year for the three models: Base Model, Robust Model $k = 1$, and Robust Model $k = 2$. However, the actual operational costs will depend on each day's failure scenario. It is worth noting that the two-stage robust model provides a BEB system design that minimizes the impact of the charging station failure.
- For example, for $k = 1$, if there is no daily charging station failure, the daily operational cost is \$2021.47. However, Supplementary Figure S.2. presents the day-to-day operational costs in each failure scenario for the Robust Model

with $k = 1$. In Supplementary Table S.6, it is shown that the operational costs in the resilient model solution are related to the daily failure scenario (higher than the case of no failure). This is attributed to the updated charging schedule after realizing the daily charging station failure uncertainty. Therefore, the annual operational costs will change depending on each day's charging station failure scenario.

- To make this point clear, the Revised Manuscript is updated as follows.
- In the "A resilient BEB system configuration" Section:

"This resulting BEB system configuration is robust against any single charging station failure. During failure, the model creates a new charging schedule utilizing the remaining charging stations and ensures that all BEBs complete their scheduled trips (An example is illustrated in Extended data IV). For instance, the energy demand of Bus ID 3 is 412.8 kWh drawn from four en-route charging stations ID 4, 7, 19, and 24 in the nominal operation (163.79 kWh, 33.33 kWh, 118.52 kWh, and 97.18 kWh, respectively). In the case of the failure of any of these four charging stations, Bus ID 3 charges the same amount from the remaining three stations."

In the Robust Model ($k = 1$), the day-to-day operational costs vary from \$2,021.47 (no disruption) to \$2,069.85 (disruption of Station ID 4). Supplementary Materials V illustrates the daily operational costs of all the scenarios of one station disruption in the Robust Model ($k = 1$)."

- In Supplementary Materials V: Daily operation costs under any disruption scenarios:

Figure S.2. Daily operation costs under any disruption scenario (Robust Model, $k = 1$)

Table S.6. Daily operation cost under all disruption scenarios (Robust Model, $k = 2$)

Disrupted Stations	Daily operational cost	Disrupted Stations	Daily operational cost	Disrupted Stations	Daily operational cost	Disrupted Stations	Daily operational cost	Disrupted Stations	Daily operational cost	Disrupted Stations	Daily operational cost
No disruption	\$1,881(3, 39)		\$1,895(7, 50)		\$1,889(11, 30)		\$1,896(17, 39)		\$1,884(25, 31)		\$1,889
(1, 2)	\$1,901(3, 43)		\$1,888(8, 9)		\$1,886(11, 31)		\$1,893(17, 43)		\$1,882(25, 33)		\$1,890
(1, 3)	\$1,910(3, 47)		\$1,890(8, 10)		\$1,884(11, 33)		\$1,890(17, 47)		\$1,883(25, 35)		\$1,888
(1, 4)	\$1,917(3, 48)		\$1,888(8, 11)		\$1,890(11, 35)		\$1,908(17, 48)		\$1,882(25, 37)		\$1,886
(1, 6)	\$1,909(3, 50)		\$1,889(8, 14)		\$1,886(11, 37)		\$1,888(17, 50)		\$1,883(25, 38)		\$1,887
(1, 7)	\$1,906(4, 6)		\$1,901(8, 15)		\$1,883(11, 38)		\$1,888(18, 19)		\$1,890(25, 39)		\$1,889
(1, 8)	\$1,899(4, 7)		\$1,907(8, 16)		\$1,884(11, 39)		\$1,891(18, 21)		\$1,890(25, 43)		\$1,887
(1, 9)	\$1,900(4, 8)		\$1,893(8, 17)		\$1,883(11, 43)		\$1,888(18, 22)		\$1,884(25, 47)		\$1,888
(1, 10)	\$1,900(4, 9)		\$1,897(8, 18)		\$1,884(11, 47)		\$1,890(18, 24)		\$1,888(25, 48)		\$1,886
(1, 11)	\$1,910(4, 10)		\$1,896(8, 19)		\$1,888(11, 48)		\$1,888(18, 25)		\$1,888(25, 50)		\$1,887
(1, 14)	\$1,901(4, 11)		\$1,903(8, 21)		\$1,888(11, 50)		\$1,890(18, 27)		\$1,885(27, 28)		\$1,883
(1, 15)	\$1,897(4, 14)		\$1,895(8, 22)		\$1,883(14, 15)		\$1,885(18, 28)		\$1,883(27, 30)		\$1,886
(1, 16)	\$1,898(4, 15)		\$1,892(8, 24)		\$1,888(14, 16)		\$1,886(18, 30)		\$1,887(27, 31)		\$1,885
(1, 17)	\$1,897(4, 16)		\$1,893(8, 25)		\$1,887(14, 17)		\$1,884(18, 31)		\$1,886(27, 33)		\$1,885
(1, 18)	\$1,899(4, 17)		\$1,893(8, 27)		\$1,884(14, 18)		\$1,885(18, 33)		\$1,885(27, 35)		\$1,887
(1, 19)	\$1,904(4, 18)		\$1,900(8, 28)		\$1,882(14, 19)		\$1,890(18, 35)		\$1,885(27, 37)		\$1,883
(1, 21)	\$1,904(4, 19)		\$1,920(8, 30)		\$1,885(14, 21)		\$1,890(18, 37)		\$1,883(27, 38)		\$1,884

Disrupted Stations	Daily operational cost	Disrupted Stations	Daily operational cost	Disrupted Stations	Daily operational cost	Disrupted Stations	Daily operational cost	Disrupted Stations	Daily operational cost	Disrupted Stations	Daily operational cost
(1, 22)	\$1,899	(4, 21)	\$1,909	(8, 31)	\$1,885	(14, 22)	\$1,889	(18, 38)	\$1,885	(27, 39)	\$1,885
(1, 24)	\$1,904	(4, 22)	\$1,893	(8, 33)	\$1,885	(14, 24)	\$1,891	(18, 39)	\$1,885	(27, 43)	\$1,882
(1, 25)	\$1,906	(4, 24)	\$1,904	(8, 35)	\$1,884	(14, 25)	\$1,889	(18, 43)	\$1,884	(27, 47)	\$1,884
(1, 27)	\$1,899	(4, 25)	\$1,905	(8, 37)	\$1,882	(14, 27)	\$1,886	(18, 47)	\$1,885	(27, 48)	\$1,883
(1, 28)	\$1,897	(4, 27)	\$1,893	(8, 38)	\$1,884	(14, 28)	\$1,884	(18, 48)	\$1,883	(27, 50)	\$1,884
(1, 30)	\$1,911	(4, 28)	\$1,893	(8, 39)	\$1,885	(14, 30)	\$1,888	(18, 50)	\$1,884	(28, 30)	\$1,887
(1, 31)	\$1,901	(4, 30)	\$1,898	(8, 43)	\$1,882	(14, 31)	\$1,886	(19, 21)	\$1,895	(28, 31)	\$1,885
(1, 33)	\$1,901	(4, 31)	\$1,900	(8, 47)	\$1,884	(14, 33)	\$1,887	(19, 22)	\$1,888	(28, 33)	\$1,884
(1, 35)	\$1,903	(4, 33)	\$1,894	(8, 48)	\$1,883	(14, 35)	\$1,886	(19, 24)	\$1,893	(28, 35)	\$1,883
(1, 37)	\$1,897	(4, 35)	\$1,894	(8, 50)	\$1,883	(14, 37)	\$1,884	(19, 25)	\$1,893	(28, 37)	\$1,882
(1, 38)	\$1,900	(4, 37)	\$1,891	(9, 10)	\$1,887	(14, 38)	\$1,885	(19, 27)	\$1,888	(28, 38)	\$1,883
(1, 39)	\$1,903	(4, 38)	\$1,893	(9, 11)	\$1,892	(14, 39)	\$1,886	(19, 28)	\$1,887	(28, 39)	\$1,884
(1, 43)	\$1,897	(4, 39)	\$1,894	(9, 14)	\$1,887	(14, 43)	\$1,884	(19, 30)	\$1,891	(28, 43)	\$1,882
(1, 47)	\$1,901	(4, 43)	\$1,892	(9, 15)	\$1,884	(14, 47)	\$1,886	(19, 31)	\$1,890	(28, 47)	\$1,884
(1, 48)	\$1,897	(4, 47)	\$1,894	(9, 16)	\$1,885	(14, 48)	\$1,885	(19, 33)	\$1,889	(28, 48)	\$1,881
(1, 50)	\$1,898	(4, 48)	\$1,892	(9, 17)	\$1,884	(14, 50)	\$1,885	(19, 35)	\$1,888	(28, 50)	\$1,882
(2, 3)	\$1,894	(4, 50)	\$1,893	(9, 18)	\$1,886	(15, 16)	\$1,884	(19, 37)	\$1,887	(30, 31)	\$1,892
(2, 4)	\$1,895	(6, 7)	\$1,896	(9, 19)	\$1,890	(15, 17)	\$1,882	(19, 38)	\$1,888	(30, 33)	\$1,887
(2, 6)	\$1,889	(6, 8)	\$1,889	(9, 21)	\$1,890	(15, 18)	\$1,883	(19, 39)	\$1,889	(30, 35)	\$1,890
(2, 7)	\$1,894	(6, 9)	\$1,890	(9, 22)	\$1,885	(15, 19)	\$1,887	(19, 43)	\$1,887	(30, 37)	\$1,885
(2, 8)	\$1,885	(6, 10)	\$1,889	(9, 24)	\$1,891	(15, 21)	\$1,887	(19, 47)	\$1,889	(30, 38)	\$1,887
(2, 9)	\$1,887	(6, 11)	\$1,893	(9, 25)	\$1,889	(15, 22)	\$1,883	(19, 48)	\$1,886	(30, 39)	\$1,886
(2, 10)	\$1,885	(6, 14)	\$1,893	(9, 27)	\$1,886	(15, 24)	\$1,887	(19, 50)	\$1,888	(30, 43)	\$1,885
(2, 11)	\$1,890	(6, 15)	\$1,887	(9, 28)	\$1,884	(15, 25)	\$1,886	(21, 22)	\$1,888	(30, 47)	\$1,885
(2, 14)	\$1,887	(6, 16)	\$1,888	(9, 30)	\$1,887	(15, 27)	\$1,883	(21, 24)	\$1,894	(30, 48)	\$1,885
(2, 15)	\$1,883	(6, 17)	\$1,887	(9, 31)	\$1,886	(15, 28)	\$1,881	(21, 25)	\$1,892	(30, 50)	\$1,886
(2, 16)	\$1,885	(6, 18)	\$1,889	(9, 33)	\$1,887	(15, 30)	\$1,885	(21, 27)	\$1,889	(31, 33)	\$1,887
(2, 17)	\$1,883	(6, 19)	\$1,893	(9, 35)	\$1,886	(15, 31)	\$1,884	(21, 28)	\$1,887	(31, 35)	\$1,885
(2, 18)	\$1,886	(6, 21)	\$1,893	(9, 37)	\$1,884	(15, 33)	\$1,884	(21, 30)	\$1,890	(31, 37)	\$1,884
(2, 19)	\$1,888	(6, 22)	\$1,891	(9, 38)	\$1,886	(15, 35)	\$1,882	(21, 31)	\$1,890	(31, 38)	\$1,885
(2, 21)	\$1,889	(6, 24)	\$1,898	(9, 39)	\$1,886	(15, 37)	\$1,881	(21, 33)	\$1,890	(31, 39)	\$1,886
(2, 22)	\$1,884	(6, 25)	\$1,892	(9, 43)	\$1,885	(15, 38)	\$1,883	(21, 35)	\$1,888	(31, 43)	\$1,885
(2, 24)	\$1,890	(6, 27)	\$1,888	(9, 47)	\$1,886	(15, 39)	\$1,884	(21, 37)	\$1,887	(31, 47)	\$1,886
(2, 25)	\$1,889	(6, 28)	\$1,886	(9, 48)	\$1,884	(15, 43)	\$1,882	(21, 38)	\$1,888	(31, 48)	\$1,884
(2, 27)	\$1,885	(6, 30)	\$1,891	(9, 50)	\$1,885	(15, 47)	\$1,884	(21, 39)	\$1,889	(31, 50)	\$1,885
(2, 28)	\$1,884	(6, 31)	\$1,890	(10, 11)	\$1,889	(15, 48)	\$1,882	(21, 43)	\$1,887	(33, 35)	\$1,885
(2, 30)	\$1,887	(6, 33)	\$1,890	(10, 14)	\$1,886	(15, 50)	\$1,882	(21, 47)	\$1,888	(33, 37)	\$1,884
(2, 31)	\$1,886	(6, 35)	\$1,888	(10, 15)	\$1,882	(16, 17)	\$1,883	(21, 48)	\$1,887	(33, 38)	\$1,886
(2, 33)	\$1,894	(6, 37)	\$1,887	(10, 16)	\$1,884	(16, 18)	\$1,885	(21, 50)	\$1,887	(33, 39)	\$1,889
(2, 35)	\$1,884	(6, 38)	\$1,888	(10, 17)	\$1,884	(16, 19)	\$1,888	(22, 24)	\$1,891	(33, 43)	\$1,884
(2, 37)	\$1,883	(6, 39)	\$1,889	(10, 18)	\$1,885	(16, 21)	\$1,889	(22, 25)	\$1,887	(33, 47)	\$1,886
(2, 38)	\$1,884	(6, 43)	\$1,888	(10, 19)	\$1,889	(16, 22)	\$1,884	(22, 27)	\$1,884	(33, 48)	\$1,884
(2, 39)	\$1,886	(6, 47)	\$1,889	(10, 21)	\$1,889	(16, 24)	\$1,889	(22, 28)	\$1,882	(33, 50)	\$1,885
(2, 43)	\$1,884	(6, 48)	\$1,887	(10, 22)	\$1,884	(16, 25)	\$1,888	(22, 30)	\$1,886	(35, 37)	\$1,883
(2, 47)	\$1,885	(6, 50)	\$1,887	(10, 24)	\$1,890	(16, 27)	\$1,884	(22, 31)	\$1,885	(35, 38)	\$1,884
(2, 48)	\$1,884	(7, 8)	\$1,889	(10, 25)	\$1,888	(16, 28)	\$1,883	(22, 33)	\$1,885	(35, 39)	\$1,885
(2, 50)	\$1,884	(7, 9)	\$1,891	(10, 27)	\$1,884	(16, 30)	\$1,885	(22, 35)	\$1,884	(35, 43)	\$1,883
(3, 4)	\$1,899	(7, 10)	\$1,890	(10, 28)	\$1,883	(16, 31)	\$1,885	(22, 37)	\$1,883	(35, 47)	\$1,884
(3, 6)	\$1,894	(7, 11)	\$1,894	(10, 30)	\$1,887	(16, 33)	\$1,885	(22, 38)	\$1,884	(35, 48)	\$1,883
(3, 7)	\$1,895	(7, 14)	\$1,891	(10, 31)	\$1,886	(16, 35)	\$1,884	(22, 39)	\$1,885	(35, 50)	\$1,884
(3, 8)	\$1,888	(7, 15)	\$1,888	(10, 33)	\$1,886	(16, 37)	\$1,882	(22, 43)	\$1,883	(37, 38)	\$1,883
(3, 9)	\$1,891	(7, 16)	\$1,889	(10, 35)	\$1,884	(16, 38)	\$1,884	(22, 47)	\$1,884	(37, 39)	\$1,884
(3, 10)	\$1,890	(7, 17)	\$1,888	(10, 37)	\$1,883	(16, 39)	\$1,885	(22, 48)	\$1,883	(37, 43)	\$1,881
(3, 11)	\$1,895	(7, 18)	\$1,890	(10, 38)	\$1,884	(16, 43)	\$1,883	(22, 50)	\$1,883	(37, 47)	\$1,884
(3, 14)	\$1,890	(7, 19)	\$1,896	(10, 39)	\$1,885	(16, 47)	\$1,884	(24, 25)	\$1,892	(37, 48)	\$1,882
(3, 15)	\$1,888	(7, 21)	\$1,899	(10, 43)	\$1,883	(16, 48)	\$1,883	(24, 27)	\$1,889	(37, 50)	\$1,882
(3, 16)	\$1,891	(7, 22)	\$1,889	(10, 47)	\$1,885	(16, 50)	\$1,883	(24, 28)	\$1,888	(38, 39)	\$1,885
(3, 17)	\$1,888	(7, 24)	\$1,896	(10, 48)	\$1,883	(17, 18)	\$1,884	(24, 30)	\$1,890	(38, 43)	\$1,883
(3, 18)	\$1,890	(7, 25)	\$1,893	(10, 50)	\$1,884	(17, 19)	\$1,888	(24, 31)	\$1,890	(38, 47)	\$1,885

Disrupted Stations	Daily operational cost	Disrupted Stations	Daily operational cost	Disrupted Stations	Daily operational cost	Disrupted Stations	Daily operational cost	Disrupted Stations	Daily operational cost	Disrupted Stations	Daily operational cost
(3, 19)	\$1,894	(7, 27)	\$1,889	(11, 14)	\$1,890	(17, 21)	\$1,887	(24, 33)	\$1,890	(38, 48)	\$1,882
(3, 21)	\$1,894	(7, 28)	\$1,888	(11, 15)	\$1,888	(17, 22)	\$1,883	(24, 35)	\$1,888	(38, 50)	\$1,883
(3, 22)	\$1,889	(7, 30)	\$1,892	(11, 16)	\$1,889	(17, 24)	\$1,888	(24, 37)	\$1,887	(39, 43)	\$1,884
(3, 24)	\$1,894	(7, 31)	\$1,891	(11, 17)	\$1,887	(17, 25)	\$1,886	(24, 38)	\$1,888	(39, 47)	\$1,885
(3, 25)	\$1,893	(7, 33)	\$1,891	(11, 18)	\$1,890	(17, 27)	\$1,883	(24, 39)	\$1,890	(39, 48)	\$1,883
(3, 27)	\$1,889	(7, 35)	\$1,889	(11, 19)	\$1,895	(17, 28)	\$1,881	(24, 43)	\$1,888	(39, 50)	\$1,885
(3, 28)	\$1,887	(7, 37)	\$1,888	(11, 21)	\$1,894	(17, 30)	\$1,884	(24, 47)	\$1,889	(43, 47)	\$1,883
(3, 30)	\$1,891	(7, 38)	\$1,889	(11, 22)	\$1,888	(17, 31)	\$1,884	(24, 48)	\$1,887	(43, 48)	\$1,882
(3, 31)	\$1,890	(7, 39)	\$1,890	(11, 24)	\$1,894	(17, 33)	\$1,884	(24, 50)	\$1,888	(43, 50)	\$1,882
(3, 33)	\$1,897	(7, 43)	\$1,888	(11, 25)	\$1,893	(17, 35)	\$1,883	(25, 27)	\$1,888	(47, 48)	\$1,884
(3, 35)	\$1,890	(7, 47)	\$1,889	(11, 27)	\$1,894	(17, 37)	\$1,881	(25, 28)	\$1,886	(47, 50)	\$1,884
(3, 37)	\$1,888	(7, 48)	\$1,888	(11, 28)	\$1,889	(17, 38)	\$1,882	(25, 30)	\$1,890	(48, 50)	\$1,882
(3, 38)	\$1,889										

- In “The price of Robustness (POR)” Section after Table 6:

*“*Operational costs include the electricity ToU, demand charges, and emissions costs and are estimated in the Robust Models based on the scenario of no charging station failures. After realizing the daily charging station failure uncertainty, these costs will be changed according to the updated daily charging schedule.”*

- In the Two-stage robust model Section:

“In this two-stage RO model, the decisions of the charging stations allocation, charging unit-rated power, the number of charger poles in each station, and the battery capacity of each bus are taken in the first stage of minimization (here-and-now decisions). However, the actual operation decisions, such as the charging schedule (charging decision, charging power, and segments-specific departure battery SoC) and the bus failure decision, are taken in the second stage after the charging station perturbations have occurred via the maximization over the uncertainty set Ξ (wait-and-see decisions). It is worth noting that multiplying the daily operational costs in the objective function by the parameter δ (number of workdays) does not mean that the charging station failure $\xi \in \Xi$ will last for all the δ days. This approach is utilized as the operational costs are estimated daily (depending on the charging schedule), and the objective function is represented annually. Moreover, the two-stage optimization model relies on minimizing the impact of the worst-case scenario approach, which aligns with assuming the annual operational costs are caused by the worst-case charging station failure on all days. However, this is just the worst-case scenario. The actual annual operational costs of the obtained resilient model are estimated after the realization of each day charging station failure uncertainty.”

- In the Extended Data II Section:

“The parameter δ (number of workdays) is taken as 365. However, this does not mean the charging station failure will last for the entire year. This value is used in the solution process of the proposed two-stage robust model by multiplying the daily operational cost of the worst-case failure scenario by 365 to provide the worst-case annual operational costs. However, as mentioned in the Methods and Results, the actual annual operational costs of the resilient model are estimated after the realization of the charging station failure scenario each day.”

How was the penalty cost for a failed bus determined? Was a suitably large value chosen?

- Yes, the penalty cost of a failed bus (ρ^{pen}) is assumed with a suitable large value that ensures the algorithm will not converge to a BEB system design with even one failed bus. This is attributed to the paper's contribution of providing a resilient optimal BEB system configuration, planning, and operation model that satisfies full operation under any charging station failure scenario.
- We totally agree with the reviewer that it should be clarified in the paper. Therefore, the Revised Manuscript is updated as follows.
- In the Base model Section:

“For each bus $b \in B$, $z_{b,j}$ is a binary variable indicating if bus b will fail to operate after serving segment j due to insufficient battery state of charge (SoC) under the lower limit. A high penalty cost with a suitably large value is included, ensuring a full operation is satisfied.”

- In the Extended Data II Section:

“Moreover, the failed bus penalty cost (ρ^{pen}) is set to 1 Million dollars (a suitably large value) to ensure a full operation is satisfied. This is achieved by preventing the NC&CG algorithm from convergence until guaranteeing that there are no failed buses under any charging station failure scenario.”

A few parameters were missing and not specified in Table 11.

- Thank you for the comment.
- Noted, and Table 11 is updated.

“The temporal values of the WTT GHG emission (ρ_t^{em}) are estimated based on the hourly distribution of the electricity generation sources obtained from the Ontario Power Generation (OPG) public data with 65 \$/tCO_{2e} 2023 Carbon Pollution Price. While the electricity ToU tariff (ρ_t^{elect}) is based on the Ontario winter weekday electricity fees.”

Table 2. Input values of model parameters

Parameters	Value	Reference	Parameters	Value	Reference
ρ^{st}	\$75,000	17	g^{max}	90%	18
ρ^{ch}	300 \$/kW	19	g^{min}	20%	18
ρ^{po}	\$13,000	19	τ	0.106	19
ρ^{batt}	500\$/kWh	20	λ_1	3	21
ρ^{bus}	550,000\$/bus	20	ρ^{DC}	0.1644 \$/kW	22
η^{ch}	95%	21	T_s	$\frac{2}{60}$ hour	Model Specific
p_{po}^{max}	500	Model Specific	δ	365	Model Specific
ρ^{pen}	\$1,000,000	Model Specific	N_t^{max}	10	Model Specific
Q^{max}	700	Model Specific	A^{batt}	{100,200,...,700}	Model Specific
k	{0,1,2}	Model Specific	A^{st}	{250,500,...,1500}	Model Specific
e^{base}	1.3675 kWh/km	Estimated by Advisor		Off-peak 0.073 \$/kWh	14
e^{batt}	0.00074/ km	Estimated by Advisor		Mid-peak \$0.102 \$/kWh	
			ρ_t^{elect}	On-peak \$0.151 \$/kWh	

1.11. The authors may consider conducting additional sensitivity analysis on the adopted parameters to understand their impact on the results. This analysis would provide valuable insights and could be accompanied by thorough discussions on the implications of parameter variations.

- We would like to thank the reviewer for their valuable comments that surely enhance our work.
- Sensitivity analyses are conducted to investigate the impact of some of the key parameter variations, including the charging system costs (infrastructure and chargers), battery costs, and operational costs (electricity ToU, demand charges, and GHG emissions). The sensitivity analysis results and discussions on parameter variations’ implications are added to Supplementary Materials VII.
- In the Conclusion Section:

“Sensitivity analyses are conducted to understand the impact of parameter variations, including the charging system costs (infrastructure and chargers), battery costs, and operational costs (electricity ToU, demand charges, and GHG emissions). The results of the sensitivity analysis are detailed in Supplementary Materials VII.”

- In Supplementary Materials VII – Sensitivity analysis:

“Sensitivity analyses of some key parameters, including the cost of the charging system (infrastructure and chargers), battery cost, and operational cost (electricity ToU, demand charges, and GHG emissions), are conducted for the robust model with $k = 1$. In each case, four additional scenarios are solved. These scenarios are generated by multiplying the base cost parameter with 0.5, 0.75, 1.25, and 1.5.”

Figure S.3 illustrates the change in the total annual system costs associated with the variation in the charging system cost (from 0.5 to 1.5 times the base cost). Moreover, the BEB system configuration in each scenario is presented in Table S.11.

The results indicate that increasing the charging cost will increase the system cost, which is logical. However, both are not growing at the same rate. In other words, with a multiplied factor equal to 0.5, the charging system cost and the total system cost of a Robust model with $k = 1$ are \$255,671 and \$6,996,530, respectively. After increasing the factor to 1.5 (300%), the charging system cost and the total system cost are increased by 259,131 and \$7,313,724, respectively. The reason is the trade-off between the charging systems, battery, and operational costs.

Figure S.3. The BEB system costs under various charging system costs.

Table S.11. Results of BEB system configuration under various charging system costs (Robust Models, $k = 1$)

Factor	Price of Robustness (PoR %)	Additional cost to the Base Model (\$)	Number of buses (#) × Battery size (kWh)	Number of charging stations	Power of charger units (kW)	Number of poles (#)
0.5	1.34%	\$92,769.151	69×100 20×200 2×300	22	6×250 16×500	23
0.75	1.91%	\$133,198.279	48×100 39×200 2×300 1×500 1×600	17	7×250 10×500	19
1	3.26%	\$227,207.206	54×100 33×200 2×300 1×500 1×600	18	9×250 9×500	19
1.25	3.26%	\$229,431.065	54×100 29×200 5×300 2×500 1×600	15	4×250 11×500	16
1.5	3.56%	\$250,965.070	52×100 36×200 3×300	16	8×250 8×500	17

The same observation is associated with the variation in battery cost (Figure S.4 and Table S.12). Increasing the battery price leads to a decrease in the number of large battery sizes to achieve a minimum total annual system cost. Therefore, the optimization model seeks to deploy more charging stations and increase operational cost by considering a more frequent charging schedule. An optimal trade-off between all the system components achieves the minimum total annual system cost. The PoR is increasing with the increase of the multiplier factor to the battery price parameter. This is attributed to the fact that one of the main approaches to achieving a robust model against charging station failure is to increase BEB battery size (first-stage

variables). Therefore, with an increased battery price, the Base Model will choose lower battery capacities. However, the Robust Model will increase the selected battery capacities, leading to a larger difference between the Base Model and the Robust Model (increase the PoR).

Figure S.4. The BEB system costs under various battery costs.

Table S.12. Results of BEB system configuration under various battery costs (Robust Models, $k = 1$)

Factor	Price of Robustness (PoR %)	Additional cost to the Base Model (\$)	Number of buses (#) × Battery size (kWh)	Number of charging stations	Power of charger units (kW)	Number of poles (#)
0.5	2.80%	\$184,617.775	20×100 24×200 22×300 12×400 8×500 4×600 1×700	14	7×250 7×500	15
0.75	2.83%	\$193,714.218	46×100 37×200 5×300 2×500 1×600	17	7×250 10×500	17
1	3.26%	\$227,207.206	54×100 33×200 2×300 1×500 1×600	18	9×250 9×500	19
1.25	3.32%	\$237,154.164	66×100 22×200 1×300 1×500 1×600	19	12×250 7×500	20
1.5	3.36%	\$241,690.481	81×100 8×200 1×300 1×400	23	13×250 10×500	24

The impacts of varying the cost of operational parameters (electricity ToU, demand charges, and GHG emissions) from 0.5 to 1.5 times the base prices on total system costs and configuration are presented in Figure S.5 and Table S.13. Increasing operational costs would result in a higher total system cost. However, a trade-off emerges among system components (charging system, battery capacities, and charging schedule) to optimize total system costs under increased operational costs.

These results indicate that when the operational cost parameters increase, the strategy involves enlarging battery sizes for BEBs to reduce charging demand during on- and mid-peak periods. What truly stands out is the reduced number of charging stations as operational cost parameters

continue to rise. This reduction can be attributed to including demand charge costs, estimated individually for each allocated station. Another notable trend is the decreasing PoR as the operational cost parameters grow, in contrast to previous cases. Since system operational costs are linked to the charging schedule (second-stage variables), increasing these parameters leads to larger first-stage variables (charging system and battery sizes). Consequently, the base model becomes more robust, resulting in a smaller difference between the Robust Model and the Base Model (PoR) as the operational cost parameters increase.

Figure S.5. The BEB system costs under various operational parameters costs.

Table S.13. Results of BEB system configuration under various operational parameters costs (Robust Models, $k = 1$)

Factor	Price of Robustness (PoR %)	Additional cost to the Base Model (\$)	Number of buses (#) × Battery size (kWh)	Number of charging stations	Power of charger units (kW)	Number of poles (#)
0.5	3.81%	\$248,017.696	72×100 18×200 1×300	22	17×250 5×500	23
0.75	3.47%	\$235,610.607	77×100 14×200	22	16×250 6×500	23
1	3.26%	\$227,207.206	54×100 33×200 2×300 1×500 1×600	18	9×250 9×500	19
1.25	2.79%	\$200,311.778	43×100 43×200 3×300 1×500 1×600	17	8×250 9×500	18
1.5	2.04%	\$151,632.156	20×100 53×200 11×300 3×400 2×500 2×600	16	10×250 6×500	16

In closure, the sensitivity analyses' results indicate that no dominant variables impact the outputs of the model. This is mainly attributed to the intertwined dynamic relationships between the BEB system components."

- 1.12. In regards to the experiment of the solution algorithm, it would be beneficial for the authors to discuss why the lower bound remains relatively stable throughout the iterations. Providing an analysis or explanation for this phenomenon would enhance the understanding of the algorithm's behavior and shed light on the underlying dynamics of the problem.
- We agree with the reviewer that explaining the behaviour of the solution algorithm would enhance the understanding of the dynamics of the problem.

- The following paragraphs are added to the Supplementary Materials II – Solution algorithm convergence to provide the required discussion.

“In Figure S.1, it is obvious that the upper bound (UB) reduces significantly from the initial iteration to the final one. In comparison, the lower bound (LB) increases at a relatively lower rate. For example, in the Robust Model with $k = 1$, the difference between the initial iteration and final iteration of the LB and UB are \$227,207 and \$29,409,346, respectively. This phenomenon is attributed to the nature of the UB and LB problems.

Specifically, the objective function of the master problem (MP) in each iteration provides the LB. The MP is designed to obtain the BEB system configuration in a pre-perturbation form to handle the failure scenarios, and the final iteration solution is the resilient model design that is robust against any k charging station disruption. Therefore, the objective function increases slowly through the algorithm iterations while handling more added failure scenarios. In other words, in the MP, all the BEB system configuration variables are utilized as decision variables, including the first-stage variables (allocated charging stations, charger-rated power in each station, number of charger poles, and the BEB fleet battery capacities) and the second-stage variables (charging schedule and failed buses). Therefore, the MP has the entire flexibility of decision variables to optimize the total system costs under the included set of charging station failure scenarios in each iteration. Therefore, as a pre-perturbation design, the total system costs of the MP (LB) increase slowly by adding more failure scenarios, and the relative ratio between the last iteration objective function (Robust Model) and the initial one (Base Model) is the price of robustness (PoR). Most notably, the BEB system configuration obtained by the MP in each iteration does not include any failed buses (high penalty cost).

On the other hand, the objective function of the sub-problem (SP) in each iteration provides the UB. The SP is designed to obtain the worst-case failure scenario under the resulting BEB system design from the MP. This failure scenario will be added to the MP in the next iteration. Therefore, the SP is formulated in a maximization of the post-perturbation effect way. In other words, iteratively, the SP takes the first-stage variables (allocated charging stations, charger-rated power in each station, number of charger poles, and the BEB fleet battery capacities) from the MP as parameters and estimates the worst-case charging station failure scenario using the second-stage variables (the failure random variable, charging schedule, and failed buses). With the high penalty cost of the failed buses and the SP's maximization nature, the SP's objective function (UB) is relatively higher than the MP (LB), especially in the first iterations. After that, the MP updates the first stage variables iteratively, making the BEB system design more robust. Therefore, the SP objective function decreases dramatically until the final iteration with a full resilient model when the objective function of the MP (pre-perturbation survivable design) is approximately equal to the SP (maximum post-perturbation effect).”

2. Reviewer # 2

The topic addressed by the paper is indeed highly relevant and the robust optimization approach is well appropriate and original for dealing with disruption uncertainty in the design of BEB transit systems. The paper is well written and the mathematics are correct. I would recommend publication with the minor changes:

- We would like to express our gratitude to the reviewer for their positive comments.
- We are also delighted to see the positive perceived value of the work.
- We have addressed the reviewer's comments, which indeed enhanced the manuscript's quality.

2.1. Line 139: It is stated than "the daily operation is reduced by more than 25%". Please precise that is occurs for Stations 1 and 4 only.

- Thank you for the comment.
- We agree with the reviewer.
- The sentence is modified as suggested by the reviewer as follows: "BEB system configuration: Base Model" Section.

"In the event of a single charging station failure (designated as $r = 1$), the daily operation is reduced by up to 34.03% associated with the failure of Station ID 1. The failure of Station ID 1 or 4 reduces the service by more than 25%."

- We would like to highlight that the results are updated. Reviewer #1 recommended the addition of Peak Demand Charge in the model, and as such the results are updated for the based model, robust model with $r = 1$, and robust model with $r = 2$.
- In addition, a similar statement is corrected in the Abstract as follows.

"In our case study, a single charging station failure would lead to up to 34.03% service reduction, and two simultaneous failures would reduce the service by up to 58.18%."

2.2. Table 3 and 4 should be given also for robust models with $k = 1$ and 2. Instead, we have only in Table 7 a short summary of service reduction. The more detailed resulted should be given and discussed.

- We thank the reviewer for their comment.
- Similar tables to Tables 3 and 4 for the Robust Models with $k = 1$ and 2 are provided in (Supplementary Materials VI) with discussion.
- For the Robust Model with $k = 1$, the detailed service reduction of $r = 2$ (153 scenarios) and 3 (816 scenarios) are presented.
- For the Robust Model with $k = 2$, the detailed service reduction of $r = 3$ (5,456 scenarios) is presented.
- Furthermore, the Revised Manuscript is updated as follows.
- In the "Insights from robust BEB system design: a discussion" Section:

"The detailed service reduction of $r = 2$ and $r = 3$ for the Robust Model with $k = 1$ and $r = 3$ for the Robust Model with $k = 2$ are presented in Supplementary Materials VI."

- In Supplementary Materials VI (Service reduction of the Robust Models) Section:

"In Table S.7, the service reduction of the Robust Model with $k = 1$ under two charging stations disruption ($r = 2$) is detailed. The total number of possible failure combinations is 153 (${}^{18}C_2$), with most of them (121 combinations, $\approx 79\%$) producing zero service reduction. The remaining failure scenarios (32 combinations) could reduce the service by up to a maximum of 13.73%.

Similarly, Table S.8 shows the service reduction of the Robust Model with $k = 1$ under three charging stations disruption ($r = 3$). Overall, there are 816 possible failure combinations (${}^{18}C_3$), and only 376 ($\approx 46\%$) combinations will result in service reductions with a maximum of 22.38%.

In Table S.9, the service reduction of the Robust Model with $k = 2$ under three charging stations disruption ($r = 3$) is illustrated. From 5,456 (${}^{33}C_3$) possible failure combinations, only 50 ($\approx 0.9\%$) combinations have non-zero service reductions with a maximum of 8.92%.

Lastly Table S.10 compares the performance of the three models during disruption ($r = 1, 2, \text{ and } 3$).

Table S.7. Service reduction at $r = 2$ (Robust Model with $k = 1$)

Disrupted stations	1	3	4	7	9	10	11	14	16	19	21	24	30	31	33	35	47	72
1	NA	10.80	10.18	3.55	4.57	0.00	4.03	0.00	0.00	0.00	0.72	1.88	7.78	4.69	10.89	2.49	2.44	0.00
3			0.00	3.08	0.00	0.00	0.00	0.00	1.88	0.00	0.00	0.00	0.00	0.00	13.73	0.00	2.94	0.00
4				6.58	1.54	2.19	6.99	0.00	0.00	8.76	5.72	0.00	0.00	2.76	0.00	0.00	0.00	1.43
7					0.00	0.00	0.00	0.00	0.00	0.00	0.00	0.00	0.00	0.00	3.03	0.00	0.00	0.00
9						0.00	5.84	0.00	0.00	0.00	0.00	0.00	0.00	0.00	0.00	0.00	0.00	0.00
10							0.00	0.00	0.00	0.00	0.00	0.00	0.00	0.00	0.00	0.00	0.00	0.00
11								0.00	0.00	0.00	0.00	0.00	4.00	6.25	0.00	3.94	0.00	0.14
14									0.00	0.00	0.84	0.00	0.00	0.00	0.00	0.00	0.00	0.00
16										0.00	0.00	0.00	0.00	0.00	0.00	0.00	0.00	0.00
19											0.00	0.00	0.00	0.00	0.00	0.00	0.00	0.00
21												0.00	0.00	0.00	0.00	0.00	0.00	0.00
24													0.00	0.00	0.00	0.00	0.00	0.00
30														4.98	0.00	0.00	0.00	0.00
31															0.00	0.00	0.00	0.00
33																0.00	0.00	0.00
35																	0.00	0.00
47																		0.00
72																		NA

Table S.8. Service reduction at $r = 3$ (Robust Model with $k = 1$)

Disrupted stations	Service reduction	Disrupted stations	Service reduction	Disrupted stations	Service reduction	Disrupted stations	Service reduction	Disrupted stations	Service reduction	Disrupted stations	Service reduction
(1, 3, 4)	22.38%	(1, 10, 31)	4.68%	(1, 35, 72)	2.63%	(3, 30, 47)	2.96%	(4, 16, 21)	5.72%	(9, 11, 72)	6.06%
(1, 3, 7)	10.79%	(1, 10, 33)	10.39%	(1, 47, 72)	2.44%	(3, 31, 33)	13.67%	(4, 16, 31)	2.76%	(9, 14, 21)	0.84%
(1, 3, 9)	14.67%	(1, 10, 35)	2.60%	(3, 4, 7)	9.93%	(3, 31, 47)	2.94%	(4, 16, 72)	1.43%	(9, 30, 31)	4.95%
(1, 3, 10)	10.97%	(1, 10, 47)	2.44%	(3, 4, 9)	1.43%	(3, 33, 35)	13.83%	(4, 19, 21)	8.87%	(10, 11, 30)	4.07%
(1, 3, 11)	14.96%	(1, 11, 14)	3.98%	(3, 4, 10)	2.24%	(3, 33, 47)	13.64%	(4, 19, 24)	8.96%	(10, 11, 31)	6.47%
(1, 3, 14)	10.54%	(1, 11, 16)	3.96%	(3, 4, 11)	6.99%	(3, 33, 72)	13.62%	(4, 19, 30)	9.28%	(10, 11, 35)	3.67%
(1, 3, 16)	13.28%	(1, 11, 19)	3.89%	(3, 4, 16)	1.88%	(3, 35, 47)	2.96%	(4, 19, 31)	11.47%	(10, 11, 72)	0.14%
(1, 3, 19)	11.09%	(1, 11, 21)	3.94%	(3, 4, 19)	9.07%	(3, 47, 72)	2.97%	(4, 19, 33)	9.14%	(10, 14, 21)	0.84%
(1, 3, 21)	11.23%	(1, 11, 24)	6.02%	(3, 4, 21)	5.70%	(4, 7, 9)	8.69%	(4, 19, 35)	8.74%	(10, 30, 31)	5.27%
(1, 3, 24)	12.87%	(1, 11, 30)	13.53%	(3, 4, 31)	2.72%	(4, 7, 10)	8.82%	(4, 19, 47)	9.17%	(11, 14, 21)	0.84%
(1, 3, 30)	18.98%	(1, 11, 31)	11.75%	(3, 4, 33)	13.62%	(4, 7, 11)	13.56%	(4, 19, 72)	15.25%	(11, 14, 30)	4.03%
(1, 3, 31)	15.66%	(1, 11, 33)	15.27%	(3, 4, 47)	2.92%	(4, 7, 14)	6.58%	(4, 21, 24)	5.00%	(11, 14, 31)	6.24%
(1, 3, 33)	16.75%	(1, 11, 35)	5.93%	(3, 4, 72)	2.80%	(4, 7, 16)	6.58%	(4, 21, 30)	5.72%	(11, 14, 35)	3.71%
(1, 3, 35)	13.19%	(1, 11, 47)	6.47%	(3, 7, 9)	3.06%	(4, 7, 19)	9.57%	(4, 21, 31)	8.48%	(11, 14, 72)	0.16%
(1, 3, 47)	13.37%	(1, 11, 72)	4.03%	(3, 7, 10)	3.06%	(4, 7, 21)	6.25%	(4, 21, 33)	5.72%	(11, 16, 30)	4.01%
(1, 3, 72)	11.36%	(1, 14, 21)	1.56%	(3, 7, 11)	3.17%	(4, 7, 24)	9.19%	(4, 21, 35)	5.72%	(11, 16, 31)	6.24%
(1, 4, 7)	16.91%	(1, 14, 24)	2.26%	(3, 7, 14)	3.12%	(4, 7, 30)	6.58%	(4, 21, 47)	5.72%	(11, 16, 35)	3.69%
(1, 4, 9)	17.56%	(1, 14, 30)	7.79%	(3, 7, 16)	5.00%	(4, 7, 31)	9.34%	(4, 21, 72)	7.15%	(11, 16, 72)	0.16%
(1, 4, 10)	12.76%	(1, 14, 31)	4.37%	(3, 7, 19)	2.97%	(4, 7, 33)	9.89%	(4, 24, 31)	2.76%	(11, 19, 30)	3.98%
(1, 4, 11)	19.28%	(1, 14, 33)	10.39%	(3, 7, 21)	3.01%	(4, 7, 35)	6.58%	(4, 24, 72)	1.43%	(11, 19, 31)	6.33%
(1, 4, 14)	10.68%	(1, 14, 35)	2.37%	(3, 7, 24)	3.01%	(4, 7, 47)	6.58%	(4, 30, 31)	7.63%	(11, 19, 35)	3.87%

Disrupted stations	Service reduction	Disrupted stations	Service reduction	Disrupted stations	Service reduction	Disrupted stations	Service reduction	Disrupted stations	Service reduction	Disrupted stations	Service reduction
(1, 4, 16)	10.18%	(1, 14, 47)	2.44%	(3, 7, 30)	3.08%	(4, 7, 72)	7.88%	(4, 30, 72)	1.43%	(11, 19, 72)	0.14%
(1, 4, 19)	21.81%	(1, 16, 21)	0.72%	(3, 7, 31)	3.17%	(4, 9, 10)	3.46%	(4, 31, 33)	2.76%	(11, 21, 30)	4.09%
(1, 4, 21)	16.61%	(1, 16, 24)	1.88%	(3, 7, 33)	14.24%	(4, 9, 11)	9.62%	(4, 31, 35)	2.76%	(11, 21, 31)	6.11%
(1, 4, 24)	12.44%	(1, 16, 30)	8.13%	(3, 7, 35)	3.24%	(4, 9, 14)	1.63%	(4, 31, 47)	2.76%	(11, 21, 35)	3.89%
(1, 4, 30)	21.02%	(1, 16, 31)	4.44%	(3, 7, 47)	6.13%	(4, 9, 16)	1.51%	(4, 31, 72)	5.88%	(11, 21, 72)	0.14%
(1, 4, 31)	18.08%	(1, 16, 33)	10.32%	(3, 7, 72)	3.12%	(4, 9, 19)	9.64%	(4, 33, 72)	2.29%	(11, 24, 30)	3.98%
(1, 4, 33)	18.40%	(1, 16, 35)	2.58%	(3, 9, 11)	6.45%	(4, 9, 21)	7.81%	(4, 35, 72)	1.43%	(11, 24, 31)	6.00%
(1, 4, 35)	13.26%	(1, 16, 47)	2.44%	(3, 9, 16)	1.86%	(4, 9, 24)	1.90%	(4, 47, 72)	1.43%	(11, 24, 35)	4.10%
(1, 4, 47)	12.61%	(1, 19, 21)	0.72%	(3, 9, 33)	13.65%	(4, 9, 30)	1.51%	(7, 9, 11)	5.73%	(11, 24, 72)	0.14%
(1, 4, 72)	16.47%	(1, 19, 24)	1.88%	(3, 9, 47)	2.92%	(4, 9, 31)	4.53%	(7, 9, 33)	3.05%	(11, 30, 31)	7.10%
(1, 7, 9)	7.36%	(1, 19, 30)	7.90%	(3, 10, 16)	1.86%	(4, 9, 33)	1.43%	(7, 10, 33)	2.96%	(11, 30, 33)	3.98%
(1, 7, 10)	3.55%	(1, 19, 31)	4.80%	(3, 10, 33)	13.58%	(4, 9, 35)	1.43%	(7, 11, 30)	4.03%	(11, 30, 35)	7.79%
(1, 7, 11)	7.36%	(1, 19, 33)	10.59%	(3, 10, 47)	2.94%	(4, 9, 47)	1.68%	(7, 11, 31)	6.07%	(11, 30, 47)	4.23%
(1, 7, 14)	3.60%	(1, 19, 35)	2.40%	(3, 11, 16)	1.88%	(4, 9, 72)	3.55%	(7, 11, 33)	3.03%	(11, 30, 72)	3.98%
(1, 7, 16)	3.74%	(1, 19, 47)	2.44%	(3, 11, 30)	3.96%	(4, 10, 11)	9.01%	(7, 11, 35)	3.83%	(11, 31, 33)	6.29%
(1, 7, 19)	3.57%	(1, 19, 72)	1.33%	(3, 11, 31)	6.06%	(4, 10, 14)	2.13%	(7, 11, 72)	0.16%	(11, 31, 35)	10.25%
(1, 7, 21)	3.85%	(1, 21, 24)	2.53%	(3, 11, 33)	13.81%	(4, 10, 16)	2.06%	(7, 14, 21)	0.84%	(11, 31, 47)	6.09%
(1, 7, 24)	5.46%	(1, 21, 30)	8.55%	(3, 11, 35)	3.66%	(4, 10, 19)	10.61%	(7, 14, 33)	3.06%	(11, 31, 72)	6.07%
(1, 7, 30)	12.20%	(1, 21, 31)	4.80%	(3, 11, 47)	2.96%	(4, 10, 21)	7.94%	(7, 16, 33)	2.99%	(11, 33, 35)	3.66%
(1, 7, 31)	8.13%	(1, 21, 33)	11.22%	(3, 11, 72)	0.14%	(4, 10, 24)	2.11%	(7, 19, 24)	2.24%	(11, 33, 72)	0.16%
(1, 7, 33)	10.37%	(1, 21, 35)	2.78%	(3, 14, 16)	1.88%	(4, 10, 30)	2.24%	(7, 19, 33)	2.99%	(11, 35, 47)	3.98%
(1, 7, 35)	6.25%	(1, 21, 47)	2.60%	(3, 14, 21)	0.86%	(4, 10, 31)	2.76%	(7, 21, 33)	3.01%	(11, 35, 72)	3.71%
(1, 7, 47)	6.02%	(1, 21, 72)	0.72%	(3, 14, 33)	13.67%	(4, 10, 33)	2.24%	(7, 24, 33)	3.14%	(11, 47, 72)	0.14%
(1, 7, 72)	3.55%	(1, 24, 30)	9.82%	(3, 14, 47)	2.92%	(4, 10, 35)	2.06%	(7, 30, 31)	5.07%	(14, 16, 21)	0.84%
(1, 9, 10)	4.68%	(1, 24, 31)	6.49%	(3, 16, 19)	1.86%	(4, 10, 47)	2.08%	(7, 30, 33)	3.05%	(14, 19, 21)	0.86%
(1, 9, 11)	14.16%	(1, 24, 33)	12.63%	(3, 16, 21)	1.86%	(4, 10, 72)	3.67%	(7, 31, 33)	3.05%	(14, 21, 24)	0.86%
(1, 9, 14)	3.49%	(1, 24, 35)	4.21%	(3, 16, 24)	1.88%	(4, 11, 14)	6.99%	(7, 33, 35)	2.99%	(14, 21, 30)	0.84%
(1, 9, 16)	3.46%	(1, 24, 47)	4.28%	(3, 16, 30)	1.86%	(4, 11, 16)	6.99%	(7, 33, 47)	3.01%	(14, 21, 31)	0.86%
(1, 9, 19)	5.03%	(1, 24, 72)	1.88%	(3, 16, 31)	1.88%	(4, 11, 19)	14.82%	(7, 33, 72)	3.03%	(14, 21, 33)	0.86%
(1, 9, 21)	3.66%	(1, 30, 31)	9.34%	(3, 16, 33)	15.32%	(4, 11, 21)	12.70%	(9, 10, 11)	5.93%	(14, 21, 35)	0.07%
(1, 9, 24)	6.13%	(1, 30, 33)	18.63%	(3, 16, 35)	1.88%	(4, 11, 24)	6.99%	(9, 10, 31)	2.01%	(14, 21, 47)	0.84%
(1, 9, 30)	8.60%	(1, 30, 35)	12.72%	(3, 16, 47)	5.05%	(4, 11, 30)	10.89%	(9, 11, 14)	5.68%	(14, 21, 72)	0.88%
(1, 9, 31)	9.08%	(1, 30, 47)	10.97%	(3, 16, 72)	1.86%	(4, 11, 31)	11.32%	(9, 11, 16)	5.77%	(14, 30, 31)	5.23%
(1, 9, 33)	13.94%	(1, 30, 72)	8.06%	(3, 19, 33)	13.67%	(4, 11, 33)	6.99%	(9, 11, 19)	5.64%	(16, 30, 31)	4.98%

Disrupted stations	Service reduction	Disrupted stations	Service reduction	Disrupted stations	Service reduction	Disrupted stations	Service reduction	Disrupted stations	Service reduction	Disrupted stations	Service reduction
(1, 9, 35)	6.86%	(1, 31, 33)	14.96%	(3, 19, 47)	2.96%	(4, 11, 35)	10.55%	(9, 11, 21)	5.86%	(19, 30, 31)	4.96%
(1, 9, 47)	5.91%	(1, 31, 35)	6.81%	(3, 21, 33)	13.80%	(4, 11, 47)	6.99%	(9, 11, 24)	5.64%	(21, 30, 31)	5.39%
(1, 9, 72)	3.73%	(1, 31, 47)	6.84%	(3, 21, 47)	2.96%	(4, 11, 72)	8.58%	(9, 11, 30)	9.60%	(24, 30, 31)	5.03%
(1, 10, 11)	6.58%	(1, 31, 72)	4.41%	(3, 24, 33)	13.65%	(4, 14, 19)	8.85%	(9, 11, 31)	11.83%	(30, 31, 33)	4.93%
(1, 10, 14)	0.52%	(1, 33, 35)	12.79%	(3, 24, 47)	2.96%	(4, 14, 21)	6.59%	(9, 11, 33)	5.70%	(30, 31, 35)	5.02%
(1, 10, 21)	0.72%	(1, 33, 47)	16.20%	(3, 30, 31)	5.30%	(4, 14, 31)	2.76%	(9, 11, 35)	9.59%	(30, 31, 47)	5.14%
(1, 10, 24)	1.88%	(1, 33, 72)	10.41%	(3, 30, 33)	13.64%	(4, 14, 72)	1.43%	(9, 11, 47)	5.64%	(30, 31, 72)	4.91%
(1, 10, 30)	7.81%	(1, 35, 47)	5.09%	(3, 30, 35)	1.94%	(4, 16, 19)	8.51%				

Table S.9. Service reduction at $r = 3$ (Robust Model with $k = 2$)

Disrupted stations	Service reduction	Disrupted stations	Service reduction	Disrupted stations	Service reduction	Disrupted stations	Service reduction	Disrupted stations	Service reduction	Disrupted stations	Service reduction
(1, 2, 3)	3.58%	(1, 11, 27)	3.08%	(1, 30, 38)	3.53%	(3, 33, 39)	0.84%	(4, 11, 31)	2.10%	(4, 31, 50)	2.01%
(1, 2, 7)	3.60%	(1, 11, 30)	5.39%	(1, 33, 47)	0.27%	(4, 7, 18)	1.77%	(4, 17, 19)	0.43%	(6, 14, 22)	1.47%
(1, 2, 33)	8.92%	(1, 11, 31)	1.00%	(1, 47, 48)	1.06%	(4, 7, 19)	4.89%	(4, 18, 19)	0.47%	(7, 19, 21)	1.79%
(1, 3, 33)	4.16%	(1, 11, 35)	3.12%	(2, 3, 7)	3.05%	(4, 7, 21)	4.96%	(4, 18, 21)	4.55%	(11, 27, 35)	2.99%
(1, 3, 39)	3.14%	(1, 22, 24)	1.09%	(2, 3, 33)	8.40%	(4, 9, 10)	1.72%	(4, 19, 21)	5.21%	(11, 30, 31)	4.39%
(1, 3, 47)	0.25%	(1, 27, 35)	1.00%	(2, 7, 33)	3.05%	(4, 9, 43)	1.77%	(4, 19, 24)	3.28%	(11, 30, 35)	1.52%
(1, 4, 6)	5.52%	(1, 28, 30)	4.23%	(3, 15, 16)	1.88%	(4, 10, 31)	2.06%	(4, 19, 25)	4.28%	(15, 16, 17)	1.83%
(1, 4, 25)	6.13%	(1, 30, 31)	4.82%	(3, 16, 17)	1.86%	(4, 10, 43)	1.13%	(4, 28, 31)	2.58%	(28, 30, 31)	4.39%
(1, 6, 30)	2.58%	(1, 30, 35)	1.94%								

Table S.10. Maximum service reductions for all models

Model	Base Model	Robust Model, $k = 1$	Robust Model, $k = 2$
Maximum service reduction if one station fails ($r = 1$)		34.03%	0.00%
Maximum service reduction if two stations fail ($r = 2$)		58.18%	13.73%
Maximum service reduction if three stations fail ($r = 3$)		59.67%	22.38%

2.3. Line 219: It is stated that the prices of robustness of 3,17% and 9,16% are marginal. Not sure 10% increase is marginal. This may be acceptable, it really depends on the context. this statement needs to be tempered.

- We totally agree with the reviewer and apologize for the inappropriate use of the word “marginal.”
- The statement is modified in the Conclusion section and also in the Abstract.
- In the Conclusion section:

“The Price of Robustness of the Robust Models with $k = 1$ and $k = 2$ are 3.26% and 8.12% higher than the base model, respectively. However, this cost prevents a significant service reduction if one (34%) or two (58%) charging stations fail. Such service reduction could lead to enormous losses in economic productivity, personal mobility, and social interaction.”

- In the Abstract:

“Our proposed two-stage robust model addresses this issue with added cost of 3.26% and 8.12% higher than the base model. However, this additional cost enables uninterrupted BEB system operation during disruption, ensuring personal mobility, social interaction, and economic productivity.”

3. References

- 1 Mukherjee, S., Nateghi, R. & Hastak, M. Data on major power outage events in the continental U.S. *Data Brief* **19**, 2079-2083, doi:10.1016/j.dib.2018.06.067 (2018).
- 2 Do, V. *et al.* Spatiotemporal distribution of power outages with climate events and social vulnerability in the USA. *Nat Commun* **14**, 2470, doi:10.1038/s41467-023-38084-6 (2023).
- 3 Raman, G., Raman, G. & Peng, J. C. Resilience of urban public electric vehicle charging infrastructure to flooding. *Nat Commun* **13**, 3213, doi:10.1038/s41467-022-30848-w (2022).
- 4 He, Y., Liu, Z. & Song, Z. Integrated charging infrastructure planning and charging scheduling for battery electric bus systems. *Transportation Research Part D: Transport and Environment* **111**, doi:10.1016/j.trd.2022.103437 (2022).
- 5 Rupp, M., Rieke, C., Handschuh, N. & Kuperjans, I. Economic and ecological optimization of electric bus charging considering variable electricity prices and CO₂eq intensities. *Transportation Research Part D: Transport and Environment* **81**, doi:10.1016/j.trd.2020.102293 (2020).
- 6 Foda, A., Abdelaty, H., Mohamed, M. & El-Sadaany, E. A generic cost-utility-emission optimization for electric bus transit infrastructure planning and charging scheduling. *Energy*, doi:10.1016/j.energy.2023.127592 (2023).
- 7 CPPSAC. Carbon pollution pricing systems across Canada. (2022).
- 8 An, K. Battery electric bus infrastructure planning under demand uncertainty. *Transportation Research Part C: Emerging Technologies* **111**, 572-587 (2020).
- 9 Weiss, M., Cloos, K. C. & Helmers, E. Energy efficiency trade-offs in small to large electric vehicles. *Environmental Sciences Europe* **32**, doi:10.1186/s12302-020-00307-8 (2020).
- 10 He, Y., Song, Z. & Liu, Z. Fast-charging station deployment for battery electric bus systems considering electricity demand charges. *Sustainable Cities and Society* **48**, doi:10.1016/j.scs.2019.101530 (2019).
- 11 Zhou, Y., Wang, H., Wang, Y. & Li, R. Robust optimization for integrated planning of electric-bus charger deployment and charging scheduling. *Transportation Research Part D: Transport and Environment* **110**, doi:10.1016/j.trd.2022.103410 (2022).
- 12 Nicolaides, D., Madhusudhanan, A. K., Na, X., Miles, J. & Cebon, D. Technoeconomic Analysis of Charging and Heating Options for an Electric Bus Service in London. *IEEE Transactions on Transportation Electrification* **5**, 769-781, doi:10.1109/tte.2019.2934356 (2019).
- 13 Abdelaty, H. & Mohamed, M. A framework for BEB energy prediction using low-resolution open-source data-driven model. *Transportation Research Part D: Transport and Environment* **103**, doi:10.1016/j.trd.2022.103170 (2022).
- 14 Foda, A., Mohamed, M. & Bakr, M. Dynamic Surrogate Trip-Level Energy Model for Electric Bus Transit System Optimization. *Transportation Research Record: Journal of the Transportation Research Board*, doi:10.1177/03611981221100242 (2022).
- 15 Gao, Z. *et al.* The Dilemma of C-Rate and Cycle Life for Lithium-Ion Batteries under Low Temperature Fast Charging. *Batteries* **8**, doi:10.3390/batteries8110234 (2022).
- 16 Abdelaty, H., Foda, A. & Mohamed, M. The Robustness of Battery Electric Bus Transit Networks under Charging Infrastructure Disruptions. *Sustainability* **15**, doi:10.3390/su15043642 (2023).

- 17 Xylia, M., Leduc, S., Patrizio, P., Kraxner, F. & Silveira, S. Locating charging infrastructure for electric buses in Stockholm. *Transportation Research Part C: Emerging Technologies* **78**, 183-200, doi:10.1016/j.trc.2017.03.005 (2017).
- 18 He, Y., Liu, Z. & Song, Z. Optimal charging scheduling and management for a fast-charging battery electric bus system. *Transportation Research Part E: Logistics and Transportation Review* **142**, doi:10.1016/j.tre.2020.102056 (2020).
- 19 Liu, Z., Song, Z. & He, Y. Economic Analysis of On-Route Fast Charging for Battery Electric Buses: Case Study in Utah. *Transportation Research Record: Journal of the Transportation Research Board* **2673**, 119-130, doi:10.1177/0361198119839971 (2019).
- 20 Quarles, N., Kockelman, K. M. & Mohamed, M. Costs and Benefits of Electrifying and Automating Bus Transit Fleets. *Sustainability* **12**, doi:10.3390/su12103977 (2020).
- 21 Liu, K., Gao, H., Wang, Y., Feng, T. & Li, C. Robust charging strategies for electric bus fleets under energy consumption uncertainty. *Transportation Research Part D: Transport and Environment* **104**, doi:10.1016/j.trd.2022.103215 (2022).
- 22 Kadri, A. & Mohammadi, F. Demand Charges Minimization for Ontario Class-A Customers Based on the Optimization of Energy Storage System. *2020 IEEE Canadian Conference on Electrical and Computer Engineering (CCECE), London, ON, Canada*, 1-4, doi:<https://doi.org/10.1109/CCECE47787.2020.9255750> (2020).

REVIEWERS' COMMENTS

Reviewer #1 (Remarks to the Author):

The authors have meticulously addressed the previous comments. As for the current manuscript:

- 1) On Page 5, Line 172, is the value of 1,805,250 kWh correct?
- 2) Is it appropriate to use the term 'recovery' or 'recovering' to describe the process of recharging electricity?

Reviewer #2 (Remarks to the Author):

I had an interesting experience in reviewing this revised manuscript. Despite the small changes made by the authors, I found the text much more readable and understandable. In fact, the various paragraphs and clarifications added help to clarify the contribution and the limitations of the study. The authors have addressed all my comments. I therefore recommend publication.

Response Letter

This response letter presents detailed discussions on our response to the comments/questions raised during the review process, along with the required references. In this respect, we present the comments of the reviewers in **blue text**, our responses and comments in black text, and the relevant revised text is shown in *italics red*. Furthermore, all modifications in the revised manuscript are highlighted in **yellow**.

1. Reviewer # 1

The authors have meticulously addressed the previous comments. As for the current manuscript:

- We are thankful to the reviewer for their time, efforts, and insightful comments, which significantly enhanced the quality of the manuscript.
- We are delighted to see that our response satisfies the reviewer's comments. We also appreciate the reviewer's kind words and positive perspective on the paper.

1.1. On Page 5, Line 172, is the value of 1,805,250 kWh, correct?

- Thank you for your comment.
- The reviewer is correct. The right value is 1,805.25 kWh.
- This value is updated in the revised manuscript as follows:

*"However, in the robust BEB system configuration with $k = 1$ and $k = 2$, the highest daily energy demands are 2,807.45 kWh ($\approx 49\%$ of the Base Model) and **1,805.25 kWh** ($\approx 32\%$ of the Base Model), respectively."*

1.2. Is it appropriate to use the term 'recovery' or 'recovering' to describe the process of recharging electricity?

- We totally agree with the reviewer that using 'recovering' will be more appropriate in describing the process of recharging electricity.
- The revised manuscript is updated according to the reviewer's suggestion.

2. Reviewer # 2

I had an interesting experience in reviewing this revised manuscript. Despite the small changes made by the authors, I found the text much more readable and understandable. In fact, the various paragraphs and clarifications added help to clarify the contribution and the limitations of the study. The authors have addressed all my comments. I therefore recommend publication.

- We are delighted by the reviewer's words.
- Indeed, the reviewer's comprehensive comments contributed substantially to the quality of the manuscript. For that, we are very grateful.